

# Random matrices and the free energy of Ising-like models with disorder

**Nils Gluth⋆, Thomas Guhr and Alfred Hucht**

Fakultät für Physik, Universität Duisburg–Essen, Duisburg, Germany

⋆ nils.gluth@uni-due.de

## Abstract

We consider an Ising model with quenched surface disorder, the disorder average of the free energy is the main object of interest. Explicit expressions for the free energy distribution are difficult to obtain if the quenched surface spins take values of ±1. Thus, we choose a different approach and model the surface disorder by Gaussian random matrices. The distribution of the free energy is calculated. We chose skew-circulant random matrices and analytically compute the characteristic function of the free energy distribution. From the characteristic function we numerically calculate the distribution and show that it becomes log-normal for sufficiently large dimensions of the disorder matrices, and in the limit of infinitely large matrices tends to a Gaussian. Furthermore, we establish a connection to the central limit theorem.

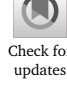

# 1 Introduction

One of the most prominent models in statistical mechanics is the anisotropic two-dimensional Ising model [1] on the $L \times M$ square lattice. Besides the exact solution of the periodic case [2,3] there has also been much work done on different boundary conditions or surface effects [4–6]. There are many interesting phenomena present in the study of the Ising model one of which is the emergence of the critical Casimir effect. It was shown by Fisher and De Gennes [7] that at the critical point the Casimir forces are universal. The critical point is the point at which the model undergoes a continuos phase transition, from a disordered high-temperature phase to an ordered low-temperature phase.

In a previous work [8] one of the present authors studied a two-dimensional Ising model of cylindrical geometry with an open boundary and another one with random surface disorder. The random surface consisted of fixed randomly orientated spins with spin values of $\pm 1$, only next neighbour interactions were included. The goal was to study the aforementioned universal properties at the critical point, mainly the behaviour of the Casimir force. The contribution to the free energy depending on the surface disorder, in the following simply referred to as free energy, was log-normal distributed for sufficiently large systems. This specific log-normal distribution tends to a Gaussian as the systems becomes arbitrarily large, however, this limit is not of interest as it goes in hand with a vanishing Casimir force.

Therefore, the goal of this paper is to investigate this log-normal behaviour to better understand its emergence. We achieve this by modelling the disorder by Gaussian random matrices, whose variance scales with the system size to the power of a disorder parameter. This requires altering the original definition of the free energy since the original definition does not make any further analytic progress permissible.

Modelling the disorder by Gaussian random matrices allows us to make use of Random Matrix Theory. Random Matrix Theory is a powerful tool originally developed to study spectral correlations. However, its scope is not limited to that. In the present work we aim to combine Random Matrix Theory and the Ising model to analyse the distribution of the free energy.

We calculate the characteristic function for the distribution of the free energy analytically and study the cumulants extensively. The distribution itself is not obtained in this work but is investigated numerically. Remarkably, this investigation shows that for all relevant values of the disorder parameter a log-normal distribution can be fitted which accurately captures the behaviour of the distribution of the free energies. Furthermore, we shows that the limit of large system sizes is Gaussian explained by the central limit theorem, resulting in constraints for the parameters of the log-normal distribution. Lastly, different intervals of the disorder parameter are identified with systems in which the disorder dominates the behaviour, an intermediate case where there is an explicit dependency on the disorder parameter in the leading order and the trivial case in which the disorder is weak enough that the system is not affected.

The present work is structured such that we first give an introduction into some key aspects of the theoretical background in Sec. 2. We then explicitly calculate the characteristic function of our system in Sec. 3 and then investigate its cumulants in Sec. 4. In Sec. 5, we will find a connection to the central limit theorem and revisit the cumulants after introducing a scaling of the variance of the disorder with the system size. These investigations are then supported by a numerical analysis in Sec. 6. In Sec. 6, we also fit the distribution with a log-normal distribution and discuss the quality of the fit. In Sec. 7, we introduce two types of similarity measures to further check the quality of the fit in Sec. 6. Lastly, in Sec. 8 we summarise our results.

# 2 Salient features of the Ising model and of random matrix theory

In Sec. 2.1 we give a brief summary of the Ising model and the specific configuration used in the present work. Furthermore, we allude to past results and formulate the starting point for our work. In Sec. 2.2 we introduce key points of Random Matrix Theory. Lastly, in Secs. 2.3 and 2.4 we give an overview of skew-circulant matrices and hypergeometric function, respectively.

## 2.1 Two-dimensional Ising model with surface disorder

The two-dimensional Ising model is one of the central models in the study of phase transitions and magnetism. It is defined on a square lattice where each lattice site is equipped with a spin. These spins interact through nearest neighbour interaction with a certain coupling constant $J$. We want to focus on the anisotropic Ising model on a cylinder with length $L$ and circumfrence $M$ at the critical point. The Hamiltonian of the system is given by

$$\mathcal{H} = -J^{\leftrightarrow} \sum_{l=1}^{L-1} \sum_{m=1}^{M} \sigma_{l,m} \sigma_{l+1,m} - J^{\updownarrow} \sum_{l=1}^{L} \sum_{m=1}^{M} \sigma_{l,m} \sigma_{l,m+1}, \tag{1}$$

with the coupling constants $J^{\leftrightarrow}$ and $J^{\updownarrow}$ in the two directions, $\sigma_{l,m}$ is the spin at the respective lattice site with values $\pm 1$. For critical systems it is useful to write the couplings as $z = \tanh(\beta J^{\leftrightarrow})$ and $t = e^{-2\beta J^{\updownarrow}}$, with the inverse temperature $\beta$. The critical point is then at

$$t = z. \tag{2}$$

In the statistical analysis the partition function,

$$Z = \operatorname{tr} \exp(-\beta \mathcal{H}) = \sum_{\{\sigma\}} \exp(-\beta \mathcal{H}), \tag{3}$$

is of interest. The trace in this case is a sum over all possible spin configurations in the system. This sum is difficult to calculate in general, since the dimension of the Hilbert space, and therefore the size of the sum, scales exponentially with the system size. However, by mapping the Ising model to a model of dimers [9] and by making use of transfer matrix methods [10] it is possible to find a much simpler expression for the case where one of the edges has open boundary conditions and the other edge has randomly chosen surface spins [8]. The resulting expression for the partition function is

$$Z = Z_1 \sqrt{\det(Q + \kappa)}, \tag{4}$$

where $Z_1$ contains bulk contributions and is independent of the effects of the random surface spins. In the Hamiltonian limit ($t = 1 = z$) [11] which means $J^{\leftrightarrow} \to \infty$ and $J^{\updownarrow} \to 0$ simultaneously while staying at the critical point, see Eq. (2), and for aspect ratio $\rho = L/M \to \infty$ it holds that

$$Q = \left[ \frac{1}{M \sin\left(\frac{\pi}{M}\left(m - n + \frac{1}{2}\right)\right)} \right]_{m,n=1,\dots,M}, \tag{5}$$

is a $M \times M$ ($M$ even) real skew-circulant matrix with all of its eigenvalues laying on the unit circle [12]. The $M \times M$ matrix,

$$\kappa = \mathrm{diag}\left(\epsilon_M \epsilon_1, \epsilon_1 \epsilon_2, \dots, \epsilon_{M-1} \epsilon_M\right), \tag{6}$$

is diagonal, where $\epsilon_i$ are the quenched surface spins with values of $\pm 1$. Quenched in this case means that the spins are fixed and therefore cannot flip in contrast to the spins in the bulk. Hence, all contributions from the disorder, the randomness, of the surface is contained in the term $\sqrt{\det(Q + \kappa)}$ of the partition function.

Another central quantity is the dimensionless free energy $F$, given in terms of the partition function as

$$F = -\ln Z = F_1 + F_2(\kappa), \quad \text{with} \quad F_1 = -\ln Z_1, \quad F_2(\kappa) = -\frac{1}{2} \ln \det(Q + \kappa). \tag{7}$$

The term $F_1$ is independent of the surface disorder and $F_2(\kappa)$ contains all contributions of the surface disorder. The free energy is particularly interesting, since it relates to important quantities like the Casimir amplitude and force [8]. However, it is crucial for this analysis to average over all possible configurations of the surface disorder

$$\left. |F_2(\kappa)| \right|_{\{\epsilon\}} \propto \sum_{\{\epsilon\}} \ln \det(Q + \kappa). \tag{8}$$

This is physically motivated since in real systems, e.g. a ferromagnet, the exact configuration is difficult to access and a statistical description is needed. Additionally, these systems are macroscopic such that the large-$M$ behaviour is particularly interesting.

Numerically it was possible to obtain a good agreement of the distribution of $F_2(\kappa)$ with an appropriately rescaled log-normal distribution [8]. Ideally, the exact distribution of $F_2(\kappa)$ in terms of the surface disorder could be obtained.

The key idea of this work is to replace the discrete disorder of $\kappa$ with a random matrix whose entries are independently Gaussian distributed. We investigate the distribution of $\ln \det(Q + \kappa)$ under the assumption that $\kappa$ is a skew-circulant matrix. Therefore, a brief summary of random matrix theory and skew-circulant matrices follows.

## 2.2 Random matrix theory

Random Matrix Theory (RMT) finds copious applications in many different areas of physics, such as quantum chaos, scattering theory, condensed matter, complex systems and wireless communication. A detailed overview of the fields of application and methodology is given in Ref. [13–15]. Random Matrix Theory is capable of modelling universal spectral statistics of large classes of systems, based on symmetries, and invariances, as well as randomness. The Hamiltonian $\mathcal{H}$ in a basis of the Hilbert space is replayed by $N \times N$ random matrices where eventually the limit $N \to \infty$ is taken. An ensemble is defined by the Gaussian probability density

$$p(H) \propto \exp\left(-\frac{1}{2v^2}\operatorname{tr} HH^\dagger\right). \tag{9}$$

Universality holds, i.e. the spectral fluctuations on the local scale of the mean level spacing do not alter when other function forms of $p(H)$ are used [13]. The most prominent ensembles are the three Dyson ensembles, the Gaussian orthogonal ensemble (GOE), the Gaussian unitary ensemble (GUE) and the Gaussian symplectic ensemble. These three ensembles are commonly characterised by the Dyson index $\overline{\beta} = 1, 2, 4$, respectively, not to be confused with the inverse temperature in Sec. 2.1. The objects of interest are the $k$-point correlation functions of the eigenvalues,

$$R_k(x_1,\ldots,x_k) = \lim_{\epsilon \to 0}\frac{1}{\pi^k}\int \mathrm{d}[H]P(H)\prod_{j=1}^{k}\operatorname{tr}\operatorname{Im}\frac{1}{\left(x_j - \iota\epsilon\right)\mathbb{1} - H}, \tag{10}$$

involving the imaginary part of the matrix resolvent, i.e. the spectral density. The volume element is the flat measure

$$\mathrm{d}[H] = \prod_{i=1}^{N}\mathrm{d}H_{nn}\prod_{n<m}\prod_{\alpha=1}^{\overline{\beta}}\mathrm{d}H_{nm}^{(\alpha)}, \tag{11}$$

where $\overline{\beta}$ is the Dyson index, if there are no symmetry constraints. For invariant integrands the diagonalisation $H = U^\dagger X U$ is useful. For $H$ being Hermitian, $U$ is a unitary matrix and $X$ is a diagonal matrix containing the eigenvalues of $H$. The volume element then transforms [16] according to

$$\mathrm{d}[H] = \Delta^2(X)\mathrm{d}[X]\mathrm{d}\mu(U), \tag{12}$$

with the Vandermonde determinant

$$\Delta(X) = \prod_{i<j}(x_j - x_i), \tag{13}$$

and the invariant Haar measure $\mathrm{d}\mu(U)$. In the present study, spectral correlations are not in the focus, but RMT will allows us to address fundamental properties of the system.

## 2.3 Skew-circulant matrices

A matrix type important for the sequel is the skew-circulant one. The defining conditions for a $M \times M$ real skew-circulant matrix $S = \left[S_{ij}\right]_{i,j=0}^{M-1}$, with $M$ even, are that its entries $S_{ij}$ are determined by the difference of the indices, that is $S_{ij} = S_{i-j} = S_m$ for $m \in \{-M+1,\ldots,M-1\}$, and that $S_m = -S_{m-M}$ for $m \in \{1,\ldots,M-1\}$ [17]. This implies that $S$ only has $M$ independent entries and that any skew-circulant matrix $S$ has to be of the form

$$S = \sum_{m=0}^{M-1}S_m h^m, \quad \text{with} \quad h_{i,j} = \delta_{i-1,j} - \delta_{i,0}\delta_{j,M-1}. \tag{14}$$

For $M = 4$ a skew-circulant matrix is given by

$$S = \begin{bmatrix} S_0 & -S_3 & -S_2 & -S_1 \\ S_1 & S_0 & -S_3 & -S_2 \\ S_2 & S_1 & S_0 & -S_3 \\ S_3 & S_2 & S_1 & S_0 \end{bmatrix}. \tag{15}$$

Hence, all skew-circulant matrices share a common eigenbasis due to the power series structure. The eigenbasis of $h$ is given by

$$h = G\lambda_h G^\dagger, \quad \text{with} \quad G_{i,j} = \frac{1}{\sqrt{M}}\omega^{i\left(j+\frac{1}{2}\right)}, \quad \text{and} \quad \lambda_{h,i} = \omega^{i+\frac{1}{2}}, \tag{16}$$

where $\omega = \exp(\iota 2\pi/M)$ is an $M$-th root of unity and $i$ ranges from 0 to $M-1$. Consequently, the eigenbasis of any skew-circulant matrix is [18]

$$S = \sum_{m=0}^{M-1} S_m h^m = G\left(\sum_{m=0}^{M} S_m \lambda_h^m\right) G^\dagger = G\lambda_S G^\dagger, \quad \text{with} \quad \lambda_{S,i} = \sum_{m=0}^{M-1} S_m \omega^{m\left(i+\frac{1}{2}\right)}. \tag{17}$$

The eigenvalues of a skew-circulant matrix with even dimension come in complex conjugated pairs which can be seen by comparing the $i$-th and the $(M-1)-i$-th eigenvalue

$$\lambda_{S,i} = \sum_{m=0}^{M-1} a_m \exp\left(\iota\frac{2\pi}{M}\left(i+\frac{1}{2}\right)m\right), \tag{18}$$

$$\lambda_{S,M-1-i} = \sum_{m=0}^{M-1} a_m \exp\left(\iota\frac{2\pi}{M}\left(M-1-i+\frac{1}{2}\right)m\right) \tag{19}$$

$$= \sum_{m=0}^{n-1} a_m \underbrace{\exp\left(\iota\frac{2M\pi}{M}m\right)}_{=1} \exp\left(\iota\frac{2\pi}{M}\left(-i-\frac{1}{2}\right)m\right) \tag{20}$$

$$= \lambda_{S,i}^\star. \tag{21}$$

From this property it follows that the determinant of any skew-circulant matrix with even dimension is always non-negative

$$\det S = \prod_{i=0}^{M-1} \lambda_{S,i} = \prod_{i=0}^{M/2-1} \lambda_{S,i}\lambda_{S,i}^\star \geq 0. \tag{22}$$

Similarly, the trace of any skew-circulant matrix times its Hermitian conjugate is non-negative,

$$\operatorname{tr} SS^\dagger = \operatorname{tr} \lambda_S \lambda_S^\star = 2\sum_{i=0}^{M/2-1} \lambda_{S,i}\lambda_{S,i}^\star \geq 0, \tag{23}$$

and reduces to a sum of only $M/2$ terms. It follows that the integration measure for ensemble $S_R(\mathbb{R}^M)$ of skew-circulant matrices is

$$d[S] = \prod_{m=0}^{M-1} dS_m \text{ with } S_m \in \mathbb{R}, \tag{24}$$

since $S_m$ are the independent entries.

## 2.4 Kummer hypergeometric functions

The most general definition of the hypergeometric function depending on a complex variable $z$ and real parameters $a_1, \ldots, a_p$ and $b_1, \ldots, b_q$ is [19]

$$
{}_pF_q\left(\begin{matrix} a_1, \ldots, a_p \\ b_1, \ldots, b_q \end{matrix} \,\middle|\, z\right) = \sum_{j=0}^{\infty} \frac{(a_1)_j \cdots (a_p)_j}{(b_1)_j \cdots (b_q)_j} \frac{z^j}{j!}, \tag{25}
$$

under the assumption that none of the $b_i$ are non-positive integers. We recall the definition of the Pochhammer symbol

$$
(x)_n = \prod_{j=0}^{n-1}(x+j). \tag{26}
$$

Kummer's differential equation

$$
z\frac{\mathrm{d}^2 w}{\mathrm{d}z^2} + (b-z)\frac{\mathrm{d}w}{\mathrm{d}z} - aw = 0, \tag{27}
$$

has among other two solutions that are of particular interest, first the Kummer confluent hypergeometric function

$$
M(a,b,z) := {}_1F_1(a,b,z) = \sum_{j=0}^{\infty} \frac{(a)_j}{(b)_j j!} z^j, \tag{28}
$$

and second

$$
\mathbf{M}(a,b,z) := \sum_{j=0}^{\infty} \frac{(a)_j}{\Gamma(b+j) j!} z^j, \tag{29}
$$

which is Olver's confluent hypergeometric function [20, Ch. 13]. The two confluent hypergeometric functions are related to each other via

$$
M(a,b,z) = \Gamma(b)\mathbf{M}(a,b,z), \tag{30}
$$

and coincide for $b = 1, 2$, since $\Gamma(1) = 1 = \Gamma(2)$. It also immediately follows from the definition of the two confluent hypergeometric function that

$$
M(0,b,z) = 1, \qquad \text{and} \qquad \mathbf{M}(0,b,z) = \frac{1}{\Gamma(b)}, \tag{31}
$$

since only the term $j = 0$ contributes to the power series.

## 3   Skew-circulant disorder

The discrete nature of $\kappa$ entering the generating function Eq. (6) makes analytical progress difficult. Thus, in the present work $\kappa$ is replaced by a real skew-circulant matrix $S$ whose independent entries are Gaussian distributed. This choice constitutes the simplest choice since we discussed in Sec. 2.3 all skew-circulant matrices share a common eigenbasis and therefore $Q$ and $S$ can be diagonalised simultaneously. Gaussian distributed skew-circulant type of disorder is realised by Gaussian distributed spin orientations in the Fourier space. In the real space the skew-circulant nature would manifest through interaction of all $M$ spins on the surface.

However, the interaction only depends on the distance between the two spins. Our goal is to calculate the distribution of $F = \ln \det(Q + S)$. The distribution of $F$ is given by

$$p_S(F|Q,\sigma) = (2\pi\sigma^2)^{-\frac{M}{2}} \int\limits_{S_R(\mathbb{R}^M)} d[S]\delta(F - \ln\det(Q+S))\exp\left(-\frac{1}{4\sigma^2}\operatorname{tr}SS^\dagger\right), \qquad (32)$$

with the integration measure defined in Eq. (24). We replace the delta function by its Fourier-representation

$$\delta(F - \ln\det(Q+S)) = \frac{1}{2\pi}\int\limits_{-\infty}^{\infty} dk\ \exp\{-\imath k(F - \ln\det(Q+S))\}, \qquad (33)$$

and introduce the characteristic function

$$\chi_S(k|Q,\sigma) = (2\pi\sigma^2)^{-\frac{M}{2}} \int\limits_{S_R(\mathbb{R}^M)} d[S]\det{}^{\imath k}(Q+S)\exp\left(-\frac{1}{4\sigma^2}\operatorname{tr}SS^\dagger\right), \qquad (34)$$

such that

$$p_S(F|Q,\sigma) = \frac{1}{2\pi}\int\limits_{-\infty}^{\infty} dk\ \exp(-\imath kF)\chi_S(k|Q,\sigma), \qquad (35)$$

where we used that the determinant of skew-circulant matrices with even dimension is positive. The integral over the skew-circulant matrices is simplified by using that any skew-circulant matrix is diagonalised by the matrix $G$, as discussed in Sec. 2.3. The matrix $G$ is constant, $dG = 0$, and therefore

$$\operatorname{tr}dS^2 = \operatorname{tr}\left(dG\lambda_S G^\dagger + Gd\lambda_S G^\dagger + G\lambda_S dG^\dagger\right)^2 = \operatorname{tr}d\lambda_S^2, \qquad (36)$$

employing the cyclic invariance of the trace. Thus, the metric of the transformation, and therefore the Jacobian, is unity since the infinitesimal length element $\operatorname{tr}dS^2$ is invariant under transformations [21]. Hence, the integration can be rewritten in terms of the eigenvalues

$$\chi_S(k|Q,\sigma) = (2\pi\sigma^2)^{-\frac{M}{2}} \int\limits_{\mathbb{C}^{M/2}} d[\lambda_S]\det{}^{\imath k}\left(\lambda_Q + \lambda_S\right)\exp\left(-\frac{1}{4\sigma^2}\operatorname{tr}\lambda_S\lambda_S^\dagger\right), \qquad (37)$$

where the integration has to be understood in the sense that $\lambda_S$ is a diagonal matrix with entries in the complex plane. The determinant and trace are given by

$$\det\left(\lambda_Q + \lambda_S\right) = \prod_{j=0}^{M/2-1}\left(\lambda_{Q,j} + \lambda_{S,j}\right)\left(\lambda_{Q,j} + \lambda_{S,j}\right)^\star = \prod_{j=0}^{M/2-1}\left|\lambda_{Q,j} + \lambda_{S,j}\right|^2, \qquad (38)$$

$$\operatorname{tr}\lambda_S\lambda_S^\dagger = 2\sum_{j=0}^{M/2-1}\left|\lambda_{S,j}\right|^2, \qquad (39)$$

and the integral over the eigenvalues factorises

$$\chi_S(k|Q,\sigma) = \prod_{j=0}^{M/2-1}\frac{1}{2\pi\sigma^2}\int_{\mathbb{C}} d^2\lambda_{S,j}\left|\lambda_{Q,j} + \lambda_{S,j}\right|^{\imath 2k}\exp\left(-\frac{\left|\lambda_{S,j}\right|^2}{2\sigma^2}\right). \qquad (40)$$

We change variables, $\lambda_{S,j} \mapsto \lambda_{S,j} + \lambda_{Q,j}$ and find

$$\chi_S(k|Q,\sigma) = \prod_{j=0}^{M/2-1} \frac{1}{2\pi\sigma^2} \int_{\mathbb{C}} d^2\lambda_{S,j} |\lambda_{S,j}|^{\iota 2k} \exp\left(-\frac{|\lambda_{S,j} - \lambda_{Q,j}|^2}{2\sigma^2}\right). \tag{41}$$

A transformation to spherical coordinates $\lambda_{S,j} = r_{S,j} \exp(\iota\varphi_{S,j})$ yields

$$\chi_S(k|Q,\sigma) = \prod_{j=0}^{M/2-1} \frac{1}{2\pi\sigma^2} \int_0^\infty dr_{S,j} \int_{-\pi}^{\pi} d\varphi_{S,j} r_{S,j}^{\iota 2k+1} \exp\left(-\frac{|r_{S,j}\exp(\iota\varphi_{S,j}) - \exp(\iota\varphi_{Q,j})|^2}{2\sigma^2}\right), \tag{42}$$

$r_{Q,j} = 1$ since $Q$ only has eigenvalues on the unit circle. The argument of the exponential function is

$$|r_{S,j}\exp(\iota\varphi_{S,j}) - \exp(\iota\varphi_{Q,j})|^2 = r_{S,j}^2 + 1 - 2r_{S,j}\cos(\varphi_{S,j} - \varphi_{Q,j}), \tag{43}$$

and the angular integral yields the Bessel function $J_0$ of order zero [20]. The characteristic function reads

$$\chi_S(k|Q,\sigma) = \prod_{j=0}^{M/2-1} \frac{\exp\left(-\frac{1}{2\sigma^2}\right)}{\sigma^2} \int_0^\infty dr_{S,j} \, r_{S,j}^{\iota 2k+1} \exp\left(-\frac{r_{S,j}^2}{2\sigma^2}\right) J_0\left(-\iota\frac{r_{S,j}}{\sigma^2}\right). \tag{44}$$

We note that the integration over the angles is quite an important step as the dependence on angular contributions of the eigenvalues of $Q$ vanishes due to the invariance of the angular $\varphi_{S,j}$ integral. The angular integrals are given by a simple product and are thus the determinant of a diagonal matrix of dimension $M/2$ with the factors in Eq. (44) as entries. The characteristic function is further simplified by the change of variable $u_j = r_{S,j}^2/2\sigma^2$

$$\chi_S(k|Q,\sigma) = \prod_{j=0}^{M/2-1} \exp\left(-\frac{1}{2\sigma^2}\right) (2\sigma^2)^{\iota k} \int_0^\infty du_j \, u_j^{\iota k} \exp(-u_j) J_0\left(-2\iota\sqrt{\frac{u_j}{2\sigma^2}}\right). \tag{45}$$

Since Olver's confluent hypergeometric function [19] may be written as

$$\mathbf{M}(a,b,-z) = \frac{z^{\frac{1}{2}(1-b)}}{\Gamma(a)} \int_0^\infty du \, u^{a-\frac{1}{2}(1+b)} \exp(-u) J_{b-1}\left(2\sqrt{zu}\right), \tag{46}$$

we arrive at

$$\chi_S(k|Q,\sigma) = \left(\exp\left(-\frac{1}{2\sigma^2}\right)(2\sigma^2)^{\iota k} \Gamma(\iota k+1) \mathbf{M}\left(\iota k+1, 1, \frac{1}{2\sigma^2}\right)\right)^{\frac{M}{2}}. \tag{47}$$

We notice the connection (30) to the Kummer function. A quick check also shows that the characteristic function is properly normalised. Since

$$\mathbf{M}(1,1,z) = \sum_{j=0}^\infty \frac{(1)_j}{\Gamma(1+j)j!} z^j = \sum_{j=0}^\infty \frac{z^j}{j!} = e^z, \tag{48}$$

we have

$$\chi_S(0|Q,\sigma) = \left(\exp\left(-\frac{1}{2\sigma^2}\right)\Gamma(1)\exp\left(\frac{1}{2\sigma^2}\right)\right)^{\frac{M}{2}} = 1. \tag{49}$$

We bring the characteristic function to a more convenient form, using the identity [20]

$$\mathbf{M}(\imath k + 1, 1, z) = {}_1F_1(\imath k + 1, 1, z) = e^z {}_1F_1(-\imath k, 1, -z) = e^z \mathbf{M}(-\imath k, 1, -z), \tag{50}$$

as

$$\chi_S(k|\sigma) = \left( (2\sigma^2)^{\imath k} \Gamma(\imath k + 1) \mathbf{M}\left(-\imath k, 1, -\frac{1}{2\sigma^2}\right) \right)^{\frac{M}{2}}. \tag{51}$$

This representation of the characteristic function is better suited for numerical calculations of the distribution, due to the absorption of the exponential prefactor into the hypergeometric function. This representation of the characteristic function is the one we will use in the following.

As shown, the characteristic function does not depend on $Q$ since all eigenvalues of $Q$ are on the unit circle and therefore have radius one. We want to mention that it is also possible to calculate the characteristic function when $Q$ has arbitrary complex eigenvalues.

## 4 Cumulants and scaling functions

As discussed, an exact functional form of the distribution cannot be obtained for large $M$ and other approaches are needed. One way to further characterise the distribution is to calculate the cumulants, which are the logarithmic derivatives of the characteristic functions at $\xi = 0$, with $\xi = \imath k$

$$\kappa_j = \partial_\xi^j \ln \chi_S(\xi|\sigma) \Big|_{\xi=0} \tag{52}$$

$$= \frac{M}{2} \partial_\xi^j \left( \xi \ln(2\sigma^2) + \ln \Gamma(\xi+1) + \ln \mathbf{M}\left(-\xi, 1, -\frac{1}{2\sigma^2}\right) \right) \Big|_{\xi=0}. \tag{53}$$

Especially the first cumulant, the mean value, is of interest due to its connection to Casimir forces. Conveniently, it has a closed form. For the higher order cumulants we were unfortunately unable to obtain such a closed form but it is still possible to make statements about the asymptotic behaviour.

It is useful to introduce a multi-index notation for the derivatives. We denote the $j$-th derivative of the confluent hypergeometric function as

$$\partial_\xi^j \mathbf{M}\left(-\xi, 1, -\frac{1}{2\sigma^2}\right) =: (-1)^j \mathbf{M}^{(j,0,0)}\left(-\xi, 1, -\frac{1}{2\sigma^2}\right), \tag{54}$$

where the superscript $(j, 0, 0)$ indicates that the confluent hypergeometric function is differentiated $j$ times with respect to the first argument, the additional factor $(-1)^j$ appears due to the chain rule.

### 4.1 First cumulant

The first cumulant is a special case since an explicit functional form exists that does not involve the use of derivatives of the hypergeometric function. According to Eq. (52), the first cumulant is

$$\kappa_1 = \frac{M}{2} \left( \ln(2\sigma^2) + \psi_0(1) - \frac{\mathbf{M}^{(1,0,0)}\left(0, 1, -1/2\sigma^2\right)}{\mathbf{M}\left(0, 1, -1/2\sigma^2\right)} \right), \tag{55}$$

with the digamma function $\psi_0(z)$, which simplifies due to

$$\psi_0(1) = -\gamma, \quad \text{and} \quad \mathbf{M}\left(0, 1, -\frac{1}{2\sigma^2}\right) = 1. \tag{56}$$

We refer to A for an explicit calculation of the first cumulant and only show the result of this calculation here

$$\kappa_1 = \frac{M}{2}\Gamma\left(0, \frac{1}{2\sigma^2}\right),\tag{57}$$

where $\Gamma(a, x)$ is the incomplete Gamma function [22].

This is a remarkably simple expression since the first cumulant of the distribution is up to a prefactor of $M/2$ given by the incomplete Gamma function $\Gamma\left(0, 1/(2\sigma^2)\right)$. For an analysis of the asymptotic behaviour of the first cumulant we refer to Appendix A. An analysis of the cumulants in the case where $1/2\sigma^2$ is also $M$-dependent will be carried out in Sec. 5.2.

### 4.2 Higher order cumulants

The cumulants are given by Eq. (53) implying that it is necessary to calculate higher order logarithmic derivatives of the Gamma and of the confluent hypergeometric function. The term $\xi \ln(2\sigma^2)$ does not contribute and can simply be neglected since it is only of order $\xi$. Making use of the chain rule the logarithmic derivatives of the Gamma function involves the polygamma-function $\psi_{j-1}$, and we observe

$$\partial_\xi^j \ln\Gamma(\xi+1) = \psi_{j-1}(\xi+1).\tag{58}$$

The logarithmic derivatives of the hypergeometric function are more involved and given in terms of derivatives of the hypergeometric functions by using Faà di Bruno's formula [23],

$$\frac{\partial^j}{\partial z^j}\ln f(z) = !\sum_{m_1+2m_2+\ldots+jm_j=j}\frac{j!}{m_1!m_2!\cdots m_j!}\frac{(-1)^{m_1+\ldots+m_j-1}(m_1+\ldots+m_j-1)!}{f(z)^{m_1+\ldots+m_j}}\prod_{l=1}^{j}\left(\frac{\partial_z^l f(z)}{l!}\right)^{m_l}.\tag{59}$$

This translates in the present case into

$$\partial_\xi^j \ln\mathbf{M}\left(-\xi, 1, -\frac{1}{2\sigma^2}\right)\bigg|_{\xi=0} = \sum_{m_1+2m_2+\ldots+jm_j=j}\frac{j!}{m_1!m_2!\cdots m_j!}(-1)^{m_1+\ldots+m_j-1}\tag{60}$$

$$\times (m_1+\ldots+m_j-1)!\prod_{l=1}^{j}\left(\frac{(-1)^l \mathbf{M}^{(l,0,0)}\left(-\xi, 1, -\frac{1}{2\sigma^2}\right)}{l!}\right)^{m_l}.$$

Consequently, the cumulants for $j > 1$ are

$$\kappa_j = \frac{M}{2}\left(\psi_{j-1}(1) + \sum_{m_1+2m_2+\ldots+jm_j=j}\frac{j!}{m_1!m_2!\cdots m_j!}(-1)^{m_1+\ldots+m_j-1}\right.$$

$$\left.\times (m_1+\ldots+m_j-1)!\prod_{l=1}^{j}\left(\frac{(-1)^l \mathbf{M}^{(l,0,0)}\left(0, 1, -\frac{1}{2\sigma^2}\right)}{l!}\right)^{m_l}\right).\tag{61}$$

We investigate the asymptotic behaviour in Appendix B.

## 5 Scaling for $M$ dependent $1/(2\sigma^2)$

For better understanding the role that the variance of the distribution of $S$ has, it is useful to substitute $S \mapsto \sqrt{2\sigma^2}^{-1}S$ in the original definition Eq. (32) of the distribution,

$$p_S(F|Q, \sigma) = (2\pi\sigma^2)^{-\frac{M}{2}}\int\limits_{S_R(\mathbb{R}^M)} d[S]\delta\left(F - \ln\det\left(Q + \sqrt{2\sigma^2}S\right)\right)\exp\left(-\frac{1}{2}\operatorname{tr}SS^\dagger\right).\tag{62}$$

The variance of $S$ now becomes a prefactor in the log-determinant which can be understood as a coupling strength between the surface disorder $S$ and the matrix $Q$ which describes the influence of the surface disorder on the bulk, see the discussion in Sec. 2.1. We choose the coupling strength in such a way that it scales with the system size $M$ where the exact behaviour is then determined by an exponent $\alpha$ and we investigate the specific case where $1/(2\sigma^2) = M^\alpha$. This choice is motivated by models like the famous Sherrington-Kirkpatrick-model in which the coupling strength also depends on the size of the system [24]. We leave the exponent variable to investigate differences in the behaviour depending on the choice of the exponent. Thus,

$$\chi_S(k|M^\alpha) = \left(M^{-\imath\alpha k}\Gamma(\imath k + 1)\mathbf{M}(-\imath k, 1, -M^\alpha)\right)^{\frac{M}{2}}, \tag{63}$$

is the characteristic function for $1/(2\sigma^2) = M^\alpha$.

## 5.1 Connection to the central limit theorem

It is now interesting to study the large-$M$ limit of the distribution with that scaling. As it is the case for many interesting systems in physics, this large-$M$ limit is directly connected to the central limit theorem which is best seen by rewriting Eq. (32) as

$$p_S(F|Q, M^\alpha) = (2\pi\sigma^2)^{-\frac{M}{2}} \int\limits_{S_R(\mathbb{R}^M)} \mathrm{d}[S]\delta(F - \ln\det(Q+S))\exp\left(-\frac{M^\alpha}{2}\operatorname{tr} SS^\dagger\right). \tag{64}$$

When applying the substitution $S \to S + Q$, transforming into the eigenbasis of the skew-circulant matrices and using spherical coordinates for the eigenvalues $\lambda_{S/Q,j} = r_{S/Q,j}\exp(\imath\phi_{S/Q,j})$, we have

$$p_S(F|Q, M^\alpha) = \prod_{j=0}^{M/2-1} \frac{M^\alpha}{\pi} \int\limits_0^\infty \mathrm{d}r_{S,j}\, r_{S,j} \exp\left(-M^\alpha\left(r_{S,j}^2 + r_{Q,j}^2\right)\right)\delta(F - \ln\det(\lambda_S))$$

$$\times \prod_{j=0}^{M/2-1} \int\limits_{-\pi}^{\pi} \mathrm{d}\phi_{S,j}\exp\left(2M^\alpha r_{S,j}r_{Q,j}\cos\left(\varphi_{S,j} - \varphi_{Q,j}\right)\right). \tag{65}$$

Recalling the properties of the determinant of a skew circulant matrices, see Sec. 2.3, the delta function is

$$\delta(F - \ln\det(\lambda_S)) = \delta\left(F - \sum_{j=0}^{M/2-1} \ln r_{S,j}^2\right). \tag{66}$$

Hence, the integral can be written more compactly as

$$p_S(F|Q, M^\alpha) = \prod_{j=0}^{M/2-1}\left(\int\limits_0^\infty \mathrm{d}r_{S,j}p\left(\ln r_{S,j}^2\right)\right)\delta\left(F - \sum_{j'=0}^{M/2-1}\ln r_{S,j'}^2\right), \tag{67}$$

with the distribution

$$p\left(\ln r_{S,j}^2\right) = \frac{M^\alpha}{\pi}r_{S,j}\exp\left(-M^\alpha\left(r_{S,j}^2 + r_{Q,j}^2\right)\right)\int\limits_{-\pi}^{\pi} \mathrm{d}\varphi_{S,j}\exp\left(2M^\alpha r_{S,j}r_{Q,j}\cos\left(\varphi_{S,j} - \varphi_{Q,j}\right)\right). \tag{68}$$

Thus, due to the delta function the integral is evaluated where $F$ is given as a sum of random variables $\ln r_{S,j}^2$ with the distribution $p\left(\ln r_{S,j}^2\right)$, For the case where $r_{Q,j} = 1$ all of those random variables are identically and independently distributed and the central limit theorem can be applied. The large-$M$ limit of $p_S(F|M^\alpha)$ is a Gaussian with mean $\mu_G = M\mu/2$ and variance $\sigma_G^2 = M\sigma^2/2$, where $\mu$ and $\sigma^2$ are the cumulants of the distribution in Eq. (68). However, the cumulants of this distribution are exactly given by the logarithmic derivatives of the characteristic function

$$\tilde{\chi}_S(k|M^\alpha) = M^{-\iota\alpha k}\Gamma(\iota k + 1)\mathbf{M}(-\iota k, 1, -M^\alpha)\,, \tag{69}$$

which one can see by carrying out the angular integrals in Eq. (68). This characteristic function is precisely the one obtained in Eq. (51) without the power $M/2$ since only one of the $M/2$ radial coordinates is accounted for. As in Sec. 5.2, this characteristic function gives the rescaled cumulants

$$\tilde{\kappa}_j = \frac{2}{M}\kappa_j\,. \tag{70}$$

Thus, the mean and variance of the Gaussian in the central limit theorem are given by the cumulants calculated in Sec. 5.2, i.e. $\mu_G = \kappa_1$ and $\sigma_G^2 = \kappa_2$.

We mention that the above derivation is not dependent on the choice $1/(2\sigma^2) = M^\alpha$ and therefore also holds in the more general case where the variance of the disorder does not depend on $M$. The central limit theorem is simply a consequence of the free energy being given by a sum of the logarithms of the eigenvalues of the disorder and that all eigenvalues of $Q$ are on the unit circle.

## 5.2 Cumulants for $M$ dependent variance

We start by revisiting the first and second cumulant and their scaling behaviour to distinguish between different cases for the exponent $\alpha$, it will become apparent that five qualitatively different regimes can be identified, as already mentioned in Sec. 4.1. In Sec. 5.2.1, we investigate the behaviour of the cumulants for $\alpha < 0$ and find an asymptotic expression for the first four cumulants. The same is done in Secs. 5.2.2 to 5.2.5 for the cases $\alpha = 0$, $0 < \alpha < 1$, $\alpha = 1$, $\alpha > 1$, respectively. At the end of the section we give a small synopsis.

### 5.2.1 Case of $\alpha < 0$

For small $1/(2\sigma^2)$, thus negative $\alpha$ and large $M$, the incomplete gamma function is asymptotically given by, see Sec. 4.1,

$$\Gamma(0, M^\alpha) \sim -\ln(M^\alpha) - \gamma + M^\alpha\,, \tag{71}$$

therefore the first cumulant is

$$\kappa_1 \sim \frac{M}{2}(-\gamma - \ln M^\alpha + M^\alpha)\,. \tag{72}$$

The leading order contributions are the logarithmic term $M/2\ln M^\alpha$ and the linear term $\gamma M/2$; whereas the term $M^{\alpha+1}/2$ will only contribute to sublinear growth, for $\alpha < -1$ it will even tend to zero for large $M$.

We stress that the asymptotics do not simply follow from the results of Sec. 4.1, as only the rescaled cumulants were analysed assuming that $1/(2\sigma^2)$ is independent of $M$. However, in the present case, where $1/(2\sigma^2)$ depends on $M$, higher orders can contribute because together with the prefactor $M/2$ they do not necessarily vanish, see Eq. (72).

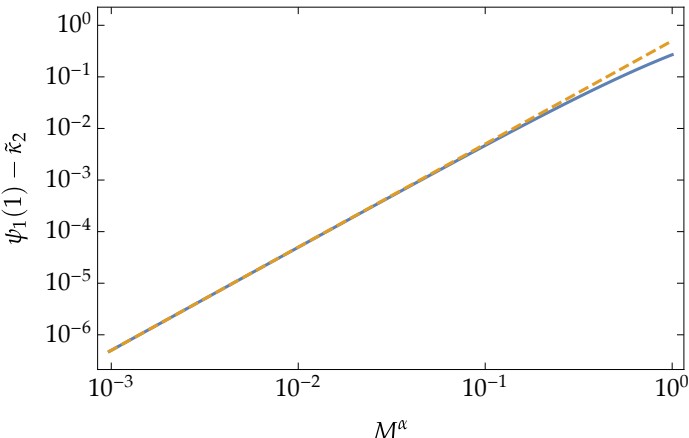

Figure 1: Scaling behaviour of $\psi_1(1)-\tilde{\kappa}_2$ versus $M^\alpha$, blue line, compared to a polynomial decay $M^{2\alpha}/2$, dashed orange line, for $\alpha < 0$.

From the above asymptotics we conclude that an important characteristic of the case $\alpha < 0$ is the leading term of the first cumulant, the mean value, is given by $M \ln M$ and we have additional linear contributions.

We adress the scaling behaviour of the higher order cumulants numerically. We do this by analysing the rescaled cumulants $\tilde{\kappa}_j$ which only depend on $M^\alpha$, as seen in Eq. (61) for $1/(2\sigma^2) = M^\alpha$, and therefore we can plot them as functions of $M^\alpha$. For $\alpha < 0$ we know that $M^\alpha \in (0,1]$ and recall that the second cumulant, the variance, converges to $\psi_1(1)M/2 = M\pi^2/12$ for small $1/(2\sigma^2)$, see Appendix B. Thus, it is interesting to analyse the quantity $\pi^2/6 - \tilde{\kappa}_2$, where $\tilde{\kappa}_2$ is the rescaled cumulant, in a double logarithmic plot, see Fig. 1.

Fig. 1 shows that the convergence behaviour is given by

$$\kappa_2 \sim \frac{M}{2}\left(\frac{\pi^2}{6} - \frac{M^{2\alpha}}{2}\right). \tag{73}$$

In leading order the variance grows linearly in $M$ with additional corrections of order $M^{1+2\alpha}$ which vanish for exponents $\alpha < -1/2$.

Consequently, the case of $\alpha < 0$ is characterised by a logarithmic growth with linear corrections in $M$ of the mean value and a linear growth in $M$ for the variance. While not necessary to distinguish between different cases it is still interesting to investigate higher order cumulants, their behaviour can be analysed numerically which we show in Figs. 2a and 2b. Hence, the higher order cumulants are asymptotically

$$\kappa_j \sim \frac{M}{2}\left(\psi_{j-1}(1) + (-1)^{j-1}c_j M^{j\alpha}\right) = \frac{M}{2}\psi_{j-1}(1) + \frac{(-1)^{j-1}}{2}c_j M^{1+j\alpha}, \tag{74}$$

where $c_j$ is some constant. This is similar to the variance where the leading contribution is linear in $M$ with additional corrections which are sublinear for $\alpha > -j^{-1}$, constant for $\alpha = -j^{-1}$ and vanish if $\alpha < -j^{-1}$. This investigation was only done for the orders two up to four but it is plausible that Eq. (74) also holds for $j > 4$, especially since we show in Appendix B that all contributions from the confluent hypergeometric function vanish in the limit of large $M$ and negative $\alpha$. This implies that the leading order of all higher order cumulants is always linear in $M$ for $\alpha < 0$.

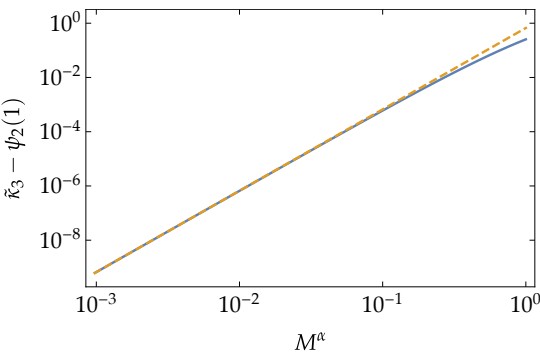

(a) Scaling behaviour of $\tilde{\kappa}_3 - \psi_2(1)$ versus $M^\alpha$, blue line, compared to a polynomial decay $\frac{2}{3}M^{3\alpha}$, dashed orange line.

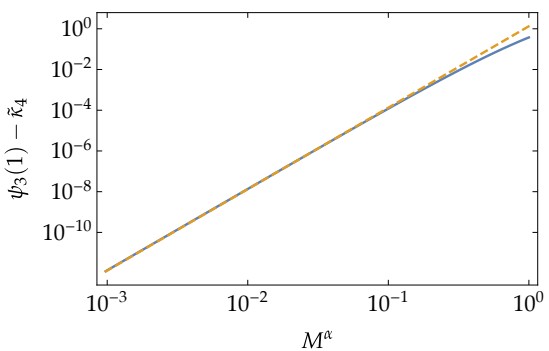

(b) Scaling behaviour of $\psi_3(1) - \tilde{\kappa}_4$ versus $M^\alpha$, blue line, compared to a polynomial decay $\frac{4}{3}M^{4\alpha}$, dashed orange line.

Figure 2: Comparison of the rescaled cumulants and polynomial decays versus $M^\alpha$ for $\alpha < 0$.

### 5.2.2 Case $\alpha = 0$

The case of $\alpha = 0$ is comparably easy since it coincides with the case of $M$ independent behaviour which was analysed in Secs. 4.1 and 4.2. Therefore, the mean and variance are respectively given by

$$\kappa_1 = \frac{M}{2}\Gamma(0,1)\,, \tag{75}$$

$$\kappa_2 = \frac{M}{2}\left(\psi_1(1) - \mathbf{M}^{(1,0,0)}(0,1,-1)^2 + \mathbf{M}^{(2,0,0)}(0,1,-1)\right)\,. \tag{76}$$

The case $\alpha = 0$ is characterised by linear growth in $M$ for mean and variance. All of the higher order cumulants depict the same behaviour as mean and variance as they grow linearly with $M$ and the slope is determined by the logarithmic derivatives of the gamma function and the confluent hypergeometric function.

### 5.2.3 Case $0 < \alpha < 1$

For $0 < \alpha < 1$ and large $M$ the asymptotic of the incomplete gamma function, found in Sec. 4.1, is given by zero. However, as before, it is necessary to analyse the exact convergence behaviour of the gamma function to determine whether the prefactor $M/2$ will yield additional corrections. This is not the case here since the incomplete Gamma function exponentially decays for large arguments

$$\Gamma(0, M^\alpha) < \exp(-M^\alpha)\,. \tag{77}$$

This implies that for large $M$ the first cumulant tends to zero

$$\kappa_1 = \frac{M}{2}\Gamma(0, M^\alpha) < \frac{M}{2}\exp(-M^\alpha) \to 0\,, \qquad \text{for } M \to \infty\,, \tag{78}$$

for all $\alpha > 0$. Thus, the case $0 < \alpha < 1$ depicts a constant mean of zero, provided that the chosen $M$ is large enough. The second cumulant is not accessible analytically and will be analysed numerically. Similar to the case $\alpha < 0$ we show the rescaled second cumulant $\tilde{\kappa}_2$ in a double logarithmically in Fig. 3 We infer that the behaviour of the second cumulant is

$$\kappa_2 \sim \frac{M}{2}2M^{-\alpha} = M^{1-\alpha}\,, \tag{79}$$

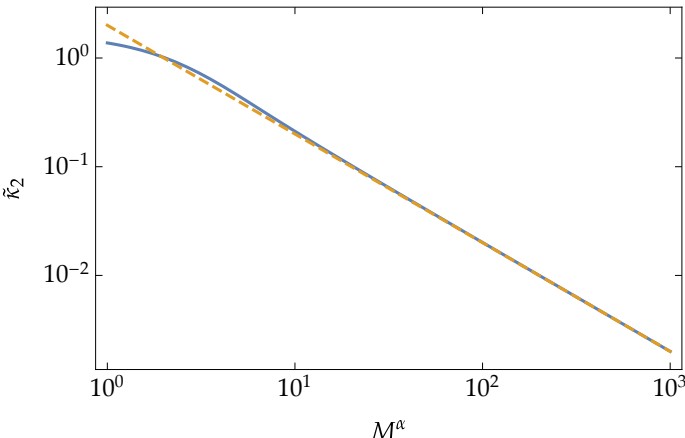

Figure 3: Scaling behaviour of $\tilde{\kappa}_2$, blue line, compared to a polynomial decay $2M^{-\alpha}$, dashed orange line, for $0 < \alpha < 1$, versus $M^\alpha$.

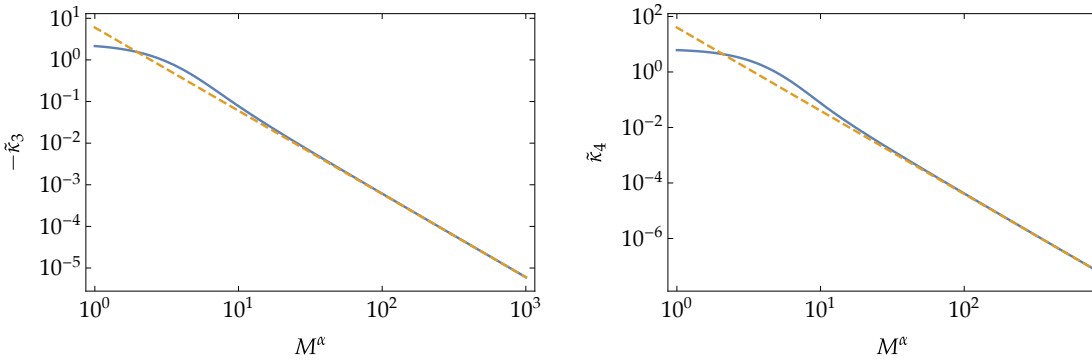

(a) Scaling behaviour of $-\tilde{\kappa}_3$ versus $M^\alpha$, blue line, compared to a polynomial decay $6M^{-2\alpha}$, dashed orange line, for $0 < \alpha < 1$.

(b) Scaling behaviour of $\tilde{\kappa}_4$ versus $M^\alpha$, blue line, compared to a polynomial decay $40M^{-3\alpha}$, dashed orange line, for $0 < \alpha < 1$.

Figure 4: Comparison between the rescaled cumulants and polynomial for $0 < \alpha < 1$, versus $M^\alpha$.

which is a sublinear growth in $M$. Thus, the characteristic behaviour in the case of $0 < \alpha < 1$ is determined by a mean of 0 and sublinear growth of the variance in $M$. To analyse the behaviour of higher order cumulants it is once again useful to look at the rescaled cumulants in a double logarithmic plot, see Figs. 4a and 4b.

The behaviour of the higher order cumulants is given by

$$\kappa_j \sim \frac{(-1)^j}{2} c'_j M^{1-(j-1)\alpha} \, , \tag{80}$$

where $c'_j$ are some constants, similar to the case $\alpha < 0$. Three possibilities exist, for $\alpha < (j-1)^{-1}$ the $j$-th cumulant depicts a sublinear growth in $M$, for $\alpha = (j-1)^{-1}$ the $j$-th cumulant is constant, and for $\alpha > (j-1)^{-1}$ the cumulant vanishes for large $M$. It is important to mention that instead of only $j$ appearing, as in the case for $\alpha < 0$, $j-1$ appears which is crucial for the behaviour of the case $\alpha = 1$ that will be dealt with in the following section. The behaviour is only shown for $j = 2, 3, 4$ but it is once again plausible that it also holds for $j > 4$.

### 5.2.4 Case $\alpha = 1$

The characteristics for $\alpha = 1$ follow directly from the above case of $0 < \alpha < 1$ since it was only assumed that $\alpha$ is greater than zero. It is still useful to distinguish the cases $\alpha = 1$ and $\alpha > 1$ from the above case, since the behaviour of the variance differs. The asymptotics of the mean value is still given by zero and the second cumulant by one which can be seen from Eq. (79) at $\alpha = 1$

$$\kappa_2 \sim 1. \tag{81}$$

As mentioned before, this results from the presence of $j-1$ instead of $j$ in Eq. (80) as was the case for $\alpha < 0$. All the higher order cumulants will converge to zero for sufficiently large $M$, see Eq. (80). Consequently, only the variance has a non-zero limit in the case $\alpha = 1$.

### 5.2.5 Case $\alpha > 1$

For $\alpha > 1$ the mean is given by zero for large $M$. However, in contrast to the case of $\alpha = 1$ the variance is not constant but rather vanishes in the limit of large $M$, following from Eq. (79) since $1 - \alpha < 0$ for $\alpha > 1$. As for $\alpha = 1$ all higher order cumulants also converge to zero for large $M$.

### 5.2.6 Synopsis

For both $\alpha = 0$ and $\alpha < 0$ the leading order contributions to the cumulants of order two or higher are linear in $M$, however the rescaled cumulants are different and for $\alpha < 0$ additional sublinear corrections appear. It is interesting that the leading order behaviour is not dependent on the value of $\alpha$, except for the first cumulant $\kappa_1$.

For the case $\alpha > 0$ we showed that the mean is exponentially suppressed and quickly converges to zero for larger $M$. For $0 < \alpha < 1$ higher order cumulants do not necessarily show that behaviour and especially the variance always grows with $M$. Unlike to the prior two cases the behaviour of the cumulants explicitly depends on the value of $\alpha$. For $\alpha = 1$ we find a constant variance while all higher order cumulants tend to zero. For $\alpha > 1$ all cumulants tend to zero.

For $\alpha > 1$ the distribution converges to a delta function. This cases is rather uninteresting since it shows that for $\alpha > 1$ the disorder in the system becomes irrelevant and therefore it will be omitted in the further investigation. Similarly we saw that for any $\alpha > 0.5$ all but the first two cumulants tend to zero which is not of particular interest either since the focus of the investigation is an intermediate limit of the distribution to capture non-Gaussian features.

## 6 Cumulant expansion and approximation of the distribution

In Sec. 6.1, we will introduce a cumulant expansion of the characteristic function and show the equivalence of the large-$M$ limit and a small $k$ approximation. In Appendix C, the approximated characteristic function is then Fourier transformed and we show that the resulting distribution verifies the Gaussian limit resulting from the central limit theorem, see Sec. 5.1. In Sec. 6.2, we present numerical data for the cumulants, compare it to the analytical expressions and find a suitable fit for the distribution. This fit and its parameters are then analysed in detail.

## 6.1 Cumulant expansion

The logarithm of the characteristic function expanded in the cumulants takes the form

$$\ln \chi_S(k|M^\alpha) = \sum_{j=1}^{\infty} \frac{\kappa_j}{j!} (\imath k)^j .$$

(82)

We notice that the exponent $M/2$ becomes a prefactor. Furthermore, it is instructive to introduce a new variable $k' = \sqrt{\kappa_2}k$ rescaled by the standard deviation. A normalising factor goes hand in hand with such a rescaling. The new cumulant generating function reads

$$\ln \chi_S(k'|M^\alpha) = -\frac{1}{2}\ln \kappa_2 + \sum_{j=1}^{\infty} \frac{1}{j!} \frac{\kappa_j}{\sqrt{\kappa_2}^j} (\imath k')^j .$$

(83)

This representation allows us to assess the relevance of the higher order cumulants on the scale of the standard deviation. We omit the cases where $\alpha > 1$ as they do not yield any particularly interesting results. The disorder in the system is not strong enough to have any significant influence. Using the scaling behaviour as calculated in Sec. 4.2 the ratio of cumulants is given by

$$\frac{\kappa_j}{\sqrt{\kappa_2}^j} \sim \begin{cases} \dfrac{(-1)^j c_j'}{2} \dfrac{M^{1-(j-1)\alpha}}{M^{j(1-\alpha)/2}} \propto M^{1-\frac{j}{2}-\alpha\left(\frac{j}{2}-1\right)}, & 0 \le \alpha \le 1, \\[4mm] 2^{\frac{j}{2}-1} \dfrac{\psi_{j-1}(1)}{\sqrt{\psi_1(1)}^j} M^{1-\frac{j}{2}}, & \alpha < 0. \end{cases}$$

(84)

For orders higher than two, the ratio tends to zero for large $M$, hence in the limit of $M \to \infty$ the cumulant generating function becomes

$$\ln \chi_S(k'|M^\alpha) \sim -\frac{1}{2}\ln \kappa_2 + \frac{\kappa_1}{\sqrt{\kappa_2}} \imath k' - \frac{1}{2}k'^2 .$$

(85)

This is the cumulant generating function of a normal distribution with mean $\kappa_1$ and variance $\kappa_2$, consistent with the application of the central limit theorem, see Sec. 5.1. However, the more important aspect is that for sufficiently large $M$ the first few cumulants produce significant contributions to the characteristic function. In this case, we choose the first four cumulants, since all higher order cumulants tend to zero with exponents larger than 2.5, implying that this drop off in the exponential function renders the contributions insignificant. The choice of which cumulants to neglect is somewhat arbitrary but it is useful to keep cumulants of up to an order larger than two to be able to observe non-Gaussian behaviour. Therefore, when comparing fits with the distribution the first few cumulants serve as a good measure to determine the accuracy of the fit.

We investigate the approximation of the distribution further in Appendix C and show that the distribution can be expanded in the cumulants which yields polynomial correction to the Gaussian. We also show that there is an equivalence between the large-$M$ and the small-$k$ limit.

## 6.2 Numerical simulations

To facilitate numerical calculations, we simplify the Fourier transform since the characteristic function has following property

$$\chi_S(-k|M^\alpha) = (\Gamma(-\imath k + 1)\mathbf{M}(\imath k, 1, -M^\alpha))^{\frac{M}{2}} = \chi_S^\star(k|M^\alpha),$$

(86)

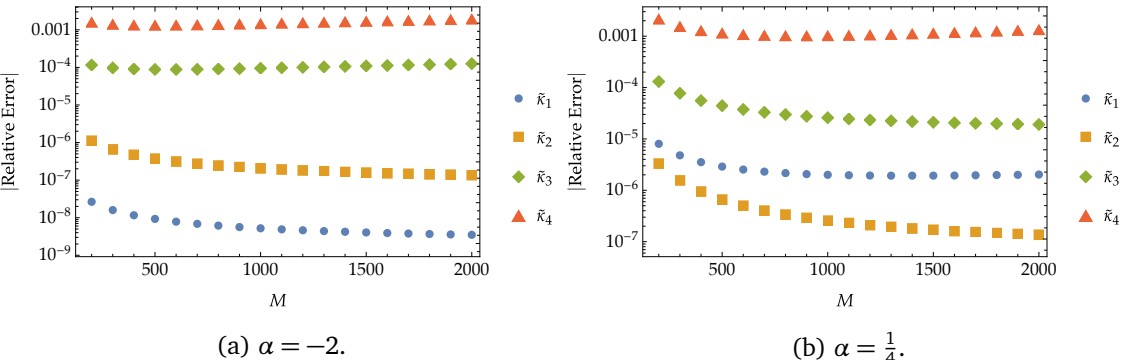

(a) $\alpha = -2$.        (b) $\alpha = \frac{1}{4}$.

Figure 5: Absolute value of the relative error between numerical and analytical results of rescaled cumulants $\kappa_i$ versus $M$ for $\alpha = -2, 1/4$.

which reflects that the distribution is real. Thus,

$$p_S(F|M^\alpha) = \frac{1}{\pi} \int_0^\infty dk \, \mathrm{Re}\left(\exp(-\iota k F) \chi_S(k|M^\alpha)\right), \tag{87}$$

the imaginary parts cancel each other.

We numerically calculate the distribution with Mathematica [25] in the interval $\left[\kappa_1 - 6\sqrt{\kappa_2}, \kappa_1 + 6\sqrt{\kappa_2}\right]$ with the standard deviation $\sqrt{\kappa_2}$. This ensures that for all combinations of $M$ and $\alpha$ the chosen intervals are comparable. The $\alpha$ values considered are $\{-2, -1.5, -1, -0.5, -0.25, 0, 0.25, 0.5\}$. For any given $\alpha$, values of $M$ range from 200 to 2000 in increments of 10.

### 6.2.1 Comparison of analytical and numerical cumulants

Higher order cumulants do not contribute significantly for larger $M$ and therefore we use the first four cumulants to determine the quality of the numerical data. In Fig. 5 we show the relative error between the analytically and numerically calculated cumulants for $\alpha = -2$ and $\alpha = 0.25$. The other values are not shown since the agreement is very good.

The comparison in Fig. 5 shows that the numerical results agree very well with the analytical ones. Some deviations exist and we attribute these to numerically inaccuracies since the largest differences are about 0.2%. Importantly, the plots show that the quantitative behaviour is captured by the numerical data. A larger interval in which the distribution is calculated is expected to improve the quality of the data, however, this comes at a cost of much larger computation times while not yielding any meaningful results and therefore it was disregarded here.

### 6.2.2 Log-normal behaviour and large-$M$ limit

It is clear from the analysis of the cumulants that the distribution is asymmetric since the third cumulant is non-zero. Another constraint is that the asymptotic for large $M$ should still be Gaussian with mean $\kappa_1$ and variance $\kappa_2$ on a standardized scale. As mentioned in Sec. 2.1 the log-normal distribution is an appropriate fit for the case of discrete disorder [8], making it a plausible choice for the case of anti-circulant disorder as well. The log-normal distribution we

used to fit the numerical data is

$$
p_{\log}(F|a, f_0, \sigma', s) = \begin{cases} \dfrac{1}{f_0} \dfrac{1}{\sqrt{2\pi\sigma'^2}\left(1 - \frac{F-s}{f_0}\right)} \exp\left(-\dfrac{\ln\left(1 - \frac{F-s}{f_0}\right)^2}{2\sigma'^2}\right), & 1 - \frac{F-s}{f_0} > 0, \\ 0, & \text{else.} \end{cases}
\tag{88}
$$

The parameter $s$ is a simple shift of $F$, which influences the mean of distribution and $f_0, \sigma'$ influence the shape and the support of the log-normal distribution. To avoid confusion we mention that the parameter $\sigma'$ is not the variance $\sigma$ of the disorder $S$ which was set to $1/(2\sigma^2) = M^\alpha$. The prefactor $f_0^{-1}$ normalizes the distribution. The parameters are determined by fitting the log-normal distribution to the numerical data. However, in the case $\alpha > 0$ we slightly modify the fit by setting $s = \kappa_1$, since we showed in Eq. (78) that the mean drops off exponentially on a scale that depends on $\alpha$ with $M$ and therefore the fit performs significantly better when fixing the parameter $s$. In Table 1, we show the scaling of the parameters determined by the fits. The leading order contributions for large $M$ indicate that for negative $\alpha$ the results are not subject to any significant change and all of the considered cases exhibit very similar behaviour. This is not surprising as for negative $\alpha$ the leading order contributions to the cumulants do not depend on $\alpha$, only going to higher orders will yield dependencies on $\alpha$. On the contrary, non-negative $\alpha$ are subject to change in the leading orders, which by a similar argument as before is not surprising, since the leading order contributions to the cumulants depend on $\alpha$. For all $\alpha > 0$ the leading order of $\sigma$ will no longer be proportional to $M^{-1/2}$ but rather some other fractional power of $M$. This is somewhat peculiar but is simply a consequence of the central limit theorem.

In the following, $\tilde{F} = F - \kappa_1$ to simplify the notation, is treated as a quantity that is on the scale of the standard deviation $\sqrt{\kappa_2}$, since the log-normal distribution only has non-zero values if $\tilde{F}/f_0 < 1$. This observation also implies that we can represent the logarithm by its power series

$$
\ln\left(1 - \frac{\tilde{F}}{f_0}\right) = \sum_{l=1}^{\infty} \frac{(-1)^{l-1}}{l}\left(-\frac{\tilde{F}}{f_0}\right)^l = -\frac{\tilde{F}}{f_0} + \mathcal{O}\left(f_0^{-2}\right),
\tag{89}
$$

where all higher order contributions of $f_0$ vanish since $f_0 \sim M$ and only the leading contributions will be considered. Additionally, since $\tilde{F}$ is of the order of the standard deviation $\sqrt{\kappa_2} \sim \sqrt{M}^{1-\Theta(\alpha)\alpha}$, $\Theta(\alpha)$ is the Heaviside step function, the factor $(1 - (F - \kappa_1)/f_0)$ is approximated by 1 with the same argument as before. Reinserting this into the log-normal distribution

Table 1: Fit parameter scaling with $M$ rounded to the fourth digit.

| $\alpha$ | $f_0$ | $\sigma'$ |
|---|---|---|
| $-2$ | $1.6882\,M + 3.7350$ | $0.5395\,M^{-\frac{1}{2}} - 0.1106\,M^{-1}$ |
| $-\frac{3}{2}$ | $1.6882\,M + 3.7348$ | $0.5395\,M^{-\frac{1}{2}} - 0.1106\,M^{-1}$ |
| $-1$ | $1.6882\,M + 3.6921$ | $0.5390\,M^{-\frac{1}{2}} - 0.1011\,M^{-1}$ |
| $-\frac{1}{2}$ | $1.6882\,M + 2.7032$ | $0.5390\,M^{-\frac{1}{2}} - 0.08916\,M^{-1}$ |
| $-\frac{1}{4}$ | $1.6735\,M - 7.5824$ | $0.5401\,M^{-\frac{1}{2}} + 0.0684\,M^{-1}$ |
| $0$ | $1.3223\,M + 2.4223$ | $0.6314\,M^{-\frac{1}{2}} - 0.1701\,M^{-1}$ |
| $\frac{1}{4}$ | $0.8261\,M - 8.6247$ | $1.3180\,M^{-\frac{5}{8}} - 1.2333\,M^{-\frac{5}{4}}$ |
| $\frac{1}{2}$ | $1.0265\,M - 16.9825$ | $0.9707\,M^{-\frac{3}{4}} + 5.1601\,M^{-\frac{3}{2}}$ |

Table 2: Leading order in $M$ of parameters compared with $\kappa_2$.

| $\alpha$ | $(f_0 \sigma')^2$ | $\kappa_2$ | relative error approx. |
|---|---|---|---|
| $-2$ | $0.829399\,M$ | $\frac{\pi^2}{12}\,M$ | 0.8% |
| $-\frac{3}{2}$ | $0.829399\,M$ | $\frac{\pi^2}{12}\,M$ | 0.8% |
| $-1$ | $0.821701\,M$ | $\frac{\pi^2}{12}\,M$ | 0.09% |
| $-\frac{1}{2}$ | $0.828087\,M$ | $\frac{\pi^2}{12}\,M$ | 0.6% |
| $-\frac{1}{4}$ | $0.817057\,M$ | $\frac{\pi^2}{12}\,M$ | 0.6% |
| $0$ | $0.697191\,M$ | $0.68794\,M$ | 1.3% |
| $\frac{1}{4}$ | $1.18549\,M^{\frac{3}{4}}$ | $M^{\frac{3}{4}}$ | 18.5% |
| $\frac{1}{2}$ | $0.992693\,\sqrt{M}$ | $\sqrt{M}$ | 0.7% |

gives

$$p_{\log}\left(F\Big|f_0, \sigma', \kappa_1\right) \sim \frac{1}{f_0\sqrt{2\pi\sigma'^2}} \exp\left(-\frac{\tilde{F}^2}{2\left(\sigma' f_0\right)^2}\right), \tag{90}$$

i.e. a Gaussian distribution with mean $\kappa_1$ and variance $\left(\sigma' f_0\right)^2$. The condition that in the large-$M$ limit the log-normal distribution converges to a Gaussian with mean $\kappa_1$ and variance $\kappa_2$ goes hand in hand with the condition that $(f\,\sigma')^2 \to \kappa_2$ for large $M$ which enforces the leading order contributions to $\sigma'$ to be $M^{-(1+\Theta(\alpha)\alpha)/2}$. This follows from the empirical result that $f_0$ depends linearly on $M$.

We investigate whether the condition $(f\,\sigma')^2 \to \kappa_2$ is indeed fulfilled from our fits and show the obtained values in Table 2.

The obtained fits accurately capture the large-$M$ limit enforced by the central limit theorem. The relative error for negative $\alpha$ is small and we attribute them to numerical errors. However, for $\alpha = 1/4$ the deviations are large which is most likely due to the values of $M$ being too small such that they are not yet able to reflect the asymptotic behaviour which was discussed previously. Nevertheless, it is remarkable how well the fits coincide with the limit for all other cases. Table 2 also shines light on why the cases for positive $\alpha$ have a different scaling for the parameter $\sigma'$, the variance no longer depends linearly on $M$ but rather with some other power. Furthermore, as $f_0$ in leading order grows linearly the scaling behaviour of $\sigma'$ changes.

Similar to the comparison between the analytically calculated cumulants and the results from the numerical data, the cumulants originating from the fitted distribution are compared to the analytically calculated cumulants and we show the relative error in Fig. 6. The examples $\alpha = -2$ and $\alpha = 1/4$ are representative for all other evaluated $\alpha$.

Just as before the cumulants calculated from the log-normal fit for negative $\alpha$ coincide very well and any deviations only appear for the 4th cumulant and are of the order of a few percent, when considering the rescaled cumulants. However, for positive $\alpha$ there are deviations for the mean which is to be expected since we fixed $s = \kappa_1$ and the first cumulant of a log-normal distribution is given by $\kappa_{\log,1} = f_0(1 - \exp(\sigma^2/2)) + s$. Similarly, the higher order cumulants show larger deviations as well, nevertheless, the qualitative behaviour is captured by the fit, see Fig. 7. We generally see that for the considered values of $\alpha$ the case $\alpha = 1/4$ performs the worst, in the range of $M$ values chosen here. We suspect that the smaller the $\alpha > 0$ the higher the $M$ values would have to be to achieve results comparable to those of negative $\alpha$. A plausible reason for this is the behaviour of the mean $\kappa_1 \sim M \exp(-M^\alpha)$ which grows linearly and then decays exponentially with growing $M$, the scale on which this decay occurs is determined by $\alpha$ and the smaller $\alpha$ the larger $M$ has to be.

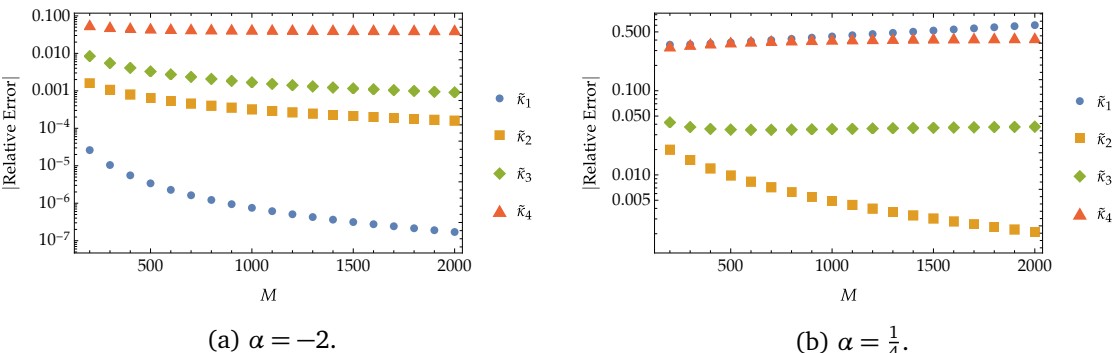

(a) $\alpha = -2$.          (b) $\alpha = \frac{1}{4}$.

Figure 6: Absolute value of the relative error between analytical results and fitted data of rescaled cumulants $\tilde{\kappa}_i$ versus $M$ for $\alpha = -2, 1/4$.

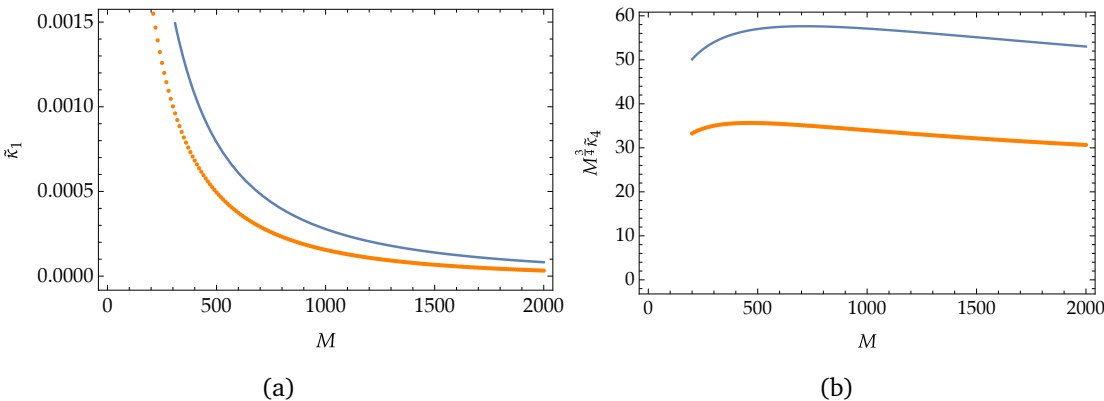

(a)           (b)

Figure 7: Rescaled cumulants $\tilde{\kappa}_1, \tilde{\kappa}_4$ versus $M$ for $\alpha = 1/4$ for analytical results (blue line) and fitted data (orange dots).

This shows that in this regime of $M$ values the log-normal distribution does capture the behaviour of the full distribution, at least up to the order that was considered here. However, the question arises whether the higher orders, which were neglected provide better understanding. Additionally, it is also interesting to study if the distribution is already converged to a Gaussian or whether log-normal behaviour is still prevalent in this regime. We give an answer to these questions by analysing the similarity between the numerical data and Gaussian/log-normal data in the following.

## 7 Similarity measures

Many measures exist to asses the quality of a fit. Here, we find it convenient to use the Jensen-Shannon divergence and Hellinger distance. The Jensen-Shannon divergence [26] for two discrete distributions $P$ and $Q$ is given by

$$\text{JSD}(P||Q) = \frac{1}{2} \left( D(P||M) + D(Q||M) \right), \quad \text{with} \quad M = \frac{1}{2}(P + Q), \tag{91}$$

where the quantity $D(P||Q)$ is the Kullback-Leibler divergence [27]

$$D(P||Q) = \sum_{i=1}^{k} p_i \ln \left( \frac{p_i}{q_i} \right), \tag{92}$$

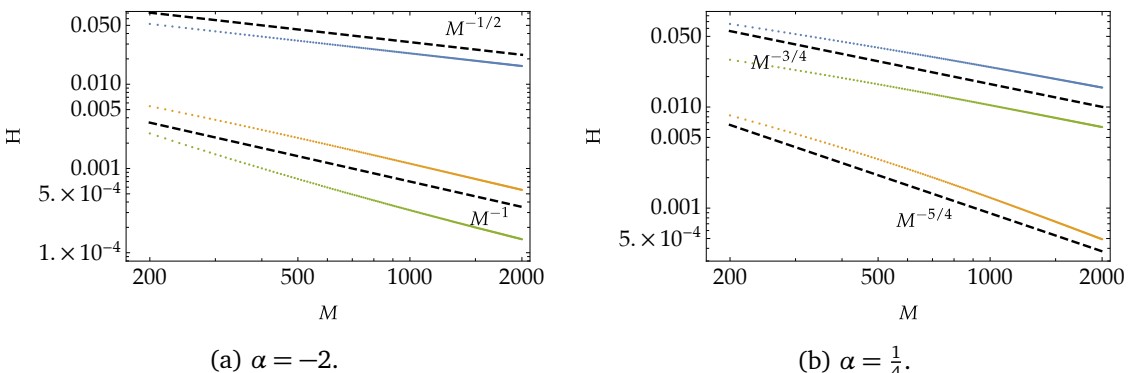

(a) $\alpha = -2$.           (b) $\alpha = \frac{1}{4}$.

Figure 8: Hellinger distance $H$ between a Gaussian (blue), the log-normal fit (green) and the fourth order approximation (orange) and the numerical data versus $M$ for different $\alpha$.

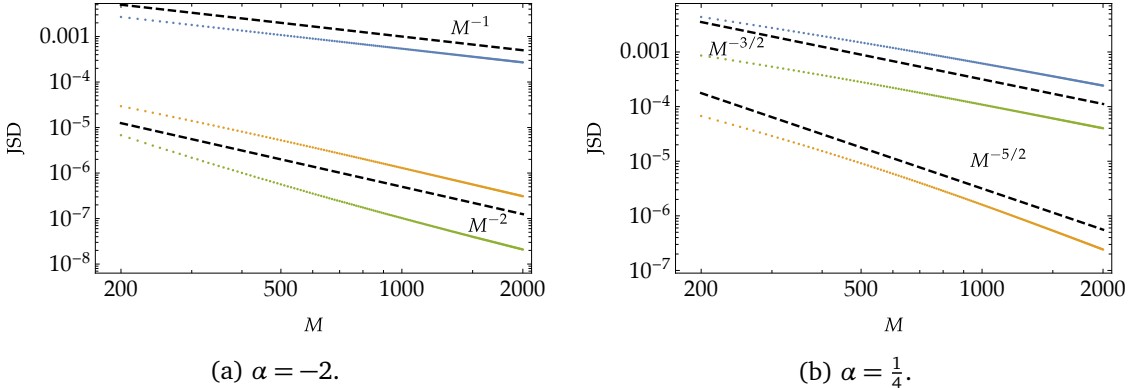

(a) $\alpha = -2$.           (b) $\alpha = \frac{1}{4}$.

Figure 9: Jensen-Shannon divergence (JSD) of a Gaussian (blue), the log-normal fit (green) and the fourth order approximation (orange) and the numerical data versus $M$ for different $\alpha$.

with $k$ the number of sample points of $P, Q$. The Kullback-Leibler divergence itself is also a valid measure for similarity however it is not symmetric in $P$ and $Q$ contrary to the Jensen-Shannon divergence.

The second measure is the Hellinger distance [28] which for discrete distributions $P, Q$ is defined by

$$H(P,Q) = \frac{1}{\sqrt{2}} \sqrt{\sum_{i=1}^{k} \left( \sqrt{p_i} - \sqrt{q_i} \right)^2}. \tag{93}$$

All of the above measures quantify the similarity between two distributions. The Kullback-Leibler and Jensen-Shannon divergence as well as the Hellinger distance indicate that two distributions are the same if the resulting value is zero. All non-zero values mean that the two distributions differ from each other. The Jensen-Shannon divergence is bounded by $\ln 2$ and the Hellinger distance by one. The Kullback-Leibler divergence is not bounded. There is no absolute scale to determine the similarity between two distributions such that we need a comparison. Therefore, we compare the similarity between the numerical data and the fitted data and the similarity between the numerical data and the Gaussian limit. This allows us to asses whether the fit captures the numerical data better than the Gaussian limit.

In Figs. 8 and 9 we show the results of these comparisons for $\alpha = -2, 1/4$. The other values are not shown because they do not differ in a meaningful way.

For both, the Hellinger distance and the Jensen-Shannon divergence, the fitted log-normal distribution provides a significantly better description of the numerically calculated data than the Gaussian limit. In the case of the Jensen-Shannon divergence it is orders of magnitude better. However, compared to a cumulant expansion of the distribution up to the fourth cumulant, the absolute value of this is used here since the approximation is not positive, for positive $\alpha$ the log-normal distribution provides a worse fit. This is in line with our prior analysis where we found qualitative agreement for positive $\alpha$ but there are quantitative differences. For all other $\alpha$'s the log-normal fit is better than the approximation. The Gaussian provides an upper bound since we know from the central limit theorem that for $M \to \infty$ our distribution is Gaussian. However, we are particularly in the behaviour deviating from the Gaussian behaviour and therefore for a reasonable fit we expect that it not only decays with $M$ but also that it is below the Gaussian curve and only meets it for $M \to \infty$.

## 8   Conclusions

We investigated the distribution of the free energy for skew-circulant disorder. We were able to obtain the characteristic function and the cumulants in a closed form. Remarkably, the first cumulant is given by the incomplete Gamma function, up to a prefactor, depending only on the variance of the disorder. We investigated the case that the variance of the disorder scales with the system size to the power of a parameter $\alpha$. We then identified different regimes in which the system is dominated by the disorder, independent of the disorder and transitional regime where the cumulants explicitly depend on $\alpha$ in the leading order. For the limit $M \to \infty$ we found that the central limit theorem holds and the distribution becomes Gaussian with mean $\kappa_1$ and variance $\kappa_2$. This limit was also accessible via a cumulant expansion and a small $k$ approximation of the characteristic function.

We computed the distribution numerically and showed that the distribution is log-normal by investigating the first four cumulants since we showed that higher cumulants become irrelevant for larger $M$. For all considered $\alpha$ we found qualitative agreement and only for positive $\alpha$ there existed significant quantitative differences. The computed parameters of the log-normal distribution also fulfilled the Gaussian limit.

Lastly, we used the Jensen-Shannon divergence and the Hellinger distance and investigated whether our log-normal fit performs better than the Gaussian limit when compared to the numerical data. This was the case for all $\alpha$. We also compared it to an approximation of the distribution up to the fourth order in the cumulants and observed that for non-positive $\alpha$ our log-normal fit was more similar to the numerical data.

In conclusion, we managed to access the characteristic function for the case of skew-circulant and its cumulants analytically. Remarkably, we also found that the distribution is well described by a log-normal fit for larger but finite $M$. Additionally, we found that in the regime where the disorder dominates the system there is no dependence on the parameter $\alpha$ and the log-normal distribution quantitatively describes the system. The transitional regime is more difficult to handle because of the explicit dependence on $\alpha$, and the log-normal distribution only captures qualitative features.

## Acknowlegdments

We are indebted to inspiring discussions with Luca Cervellera.

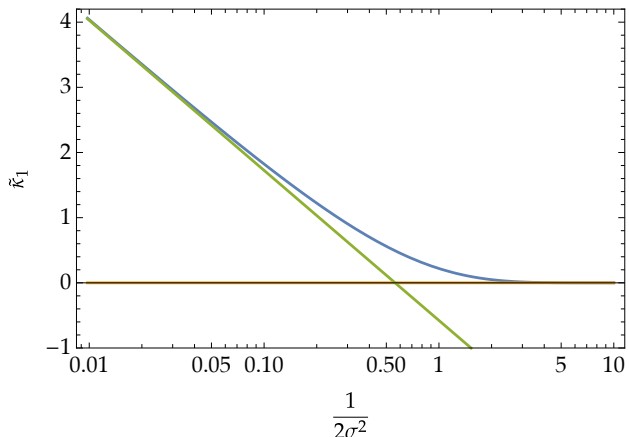

Figure 10: Rescaled first cumulant $\tilde{\kappa}_1$ versus $1/2\sigma^2$. For comparison the asymptotic expressions $-\gamma + \ln 2\sigma^2$ (green) and 0 (orange).

## A First cumulant

The derivatives of the confluent hypergeometric function, i.e. of Kummer's confluent hypergeometric function, were calculated in Ref. [29]. The first derivative,

$$\mathbf{M}^{(1,0,0)}(a,b,z) = \frac{z}{(b)_1} \sum_{m_1=0}^{\infty} \frac{(a)_{m_1}(1)_{m_1}}{(b+1)_{m_1}(2)_{m_1}} \frac{z^{m_1}}{m_1!} {}_2F_2\left( \begin{array}{c} 1, a+1+m_1 \\ 2+m_1, b+1+m_1 \end{array} \middle| z \right), \qquad \text{(A.1)}$$

reduces to a rather simple expression in the case of $a = 0, b = 1, z = -(2\sigma^2)^{-1}$,

$$\mathbf{M}^{(1,0,0)}\left(0,1,-\frac{1}{2\sigma^2}\right) = -\frac{1}{2\sigma^2} {}_2F_2\left( \begin{array}{c} 1,1 \\ 2,2 \end{array} \middle| -\frac{1}{2\sigma^2} \right). \qquad \text{(A.2)}$$

Because of the Pochhammer-symbol $(a)_{m_1}$ the sum only yields non-zero results for $m_1 = 0$. We further simplify the hypergeometric function by using the identities [30, 31] which together yield the result

$$_2F_2\left( \begin{array}{c} 1,1 \\ 2,2 \end{array} \middle| z \right) = \frac{1}{2z}\left( 2\text{Ei}(z) + \ln\frac{1}{z} - \ln z - 2\gamma \right) = \frac{-\Gamma(0,-z) - \ln(-z) - \gamma}{z}, \qquad \text{(A.3)}$$

where $\text{Ei}(z)$ is the exponential integral. Hence, the derivative of the hypergeometric function for $z = -1/(2\sigma^2)$ is given by

$$\mathbf{M}^{(1,0,0)}\left(0,1,-\frac{1}{2\sigma^2}\right) = -\Gamma\left(0,\frac{1}{2\sigma^2}\right) - \ln\frac{1}{2\sigma^2} - \gamma. \qquad \text{(A.4)}$$

This is reinserted into the expression of the first cumulant Eq. (55) together with the values from Eq. (56) to arrive at the result

$$\kappa_1 = \frac{M}{2}\left( \ln\left(2\sigma^2\right) - \gamma + \Gamma\left(0,\frac{1}{2\sigma^2}\right) - \ln\left(2\sigma^2\right) + \gamma \right) = \frac{M}{2}\Gamma\left(0,\frac{1}{2\sigma^2}\right). \qquad \text{(A.5)}$$

In Fig. 10, we plot the rescaled cumulant $\tilde{\kappa}_1$. For small values of $1/(2\sigma^2)$ the incomplete gamma function depicts a logarithmically divergent behaviour

$$\Gamma\left(0,\frac{1}{2\sigma^2}\right) \sim -\ln\left(\frac{1}{2\sigma^2}\right) - \gamma. \qquad \text{(A.6)}$$

This follows from Eq. (A.3) and inserting the definition of the exponential integral [32]

$$\mathrm{Ei}(z) = \sum_{k=1}^{\infty} \frac{z^k}{k!k} + \gamma + \frac{1}{2}\left(\ln z - \ln\frac{1}{z}\right), \tag{A.7}$$

which then gives

$$\Gamma\left(0, \frac{1}{2\sigma^2}\right) = -\sum_{k=1}^{\infty} \frac{(-1)^k}{(2\sigma^2)^k k!k} - \gamma - \ln\left(\frac{1}{2\sigma^2}\right). \tag{A.8}$$

This clearly represents the asymptotics in Eq. (A.6), as for $z = -1/2\sigma^2$ the contributions from the series become arbitrarily small for large $2\sigma^2$. In later parts of the analysis higher orders will be important and therefore we mention here that the next order is bounded linearly

$$\Gamma\left(0, \frac{1}{2\sigma^2}\right) \sim -\ln\left(\frac{1}{2\sigma^2}\right) - \gamma + \frac{1}{2\sigma^2}. \tag{A.9}$$

For large $1/2\sigma^2$ the incomplete Gamma function tends to zero since for $a = 0$ and for $2\sigma^2 < 1$ it is clear that $t^{-1} < 1$ for any $t$ in the interval and thus

$$\Gamma\left(0, \frac{1}{2\sigma^2}\right) = \int_{1/2\sigma^2}^{\infty} \mathrm{d}t \; t^{-1} \exp(-t) < \int_{1/2\sigma^2}^{\infty} \mathrm{d}t \; \exp(-t) = \exp\left(-\frac{1}{2\sigma^2}\right), \tag{A.10}$$

is bounded by an exponentially decay in $1/2\sigma^2$.

## B  Higher order cumulants

In Figs. 11a to 11c, we show the rescaled cumulants $\tilde{\kappa}_j$ for $j = 2, 3, 4$. Similar to the first cumulant it is possible to identify three regimes, one for small and one for large values of $1/(2\sigma^2)$, where the rescaled cumulants converge to a non-zero constant and zero, respectively, and a third transition area that connects both of the asymptotics. The values for the case of small $1/2\sigma^2$ coincide with those of the polygamma function $\psi_{j-1}(1)$. This is not a coincidence but rather a result of the behaviour of the derivatives of the confluent hypergeometric function. The derivatives with respect to the first argument were explicitly calculated in Ref. [29]

$$\mathbf{M}^{(j,0,0)}(a, b, z) = \frac{z^j}{(b)_j} \sum_{m_1, \ldots, m_{j+1}=0}^{\infty} \frac{(1)_{m_1} \cdots (1)_{m_{j+1}}}{(j+1)_{m_1+\ldots+m_{j+1}} (b+j)_{m_1+\ldots+m_{j+1}}}$$
$$\times \frac{(a)_{m_1} \cdots (a+j)_{m_1+\ldots+m_{j+1}}}{(a+1)_{m_1} \cdots (a+j)_{m_1+\ldots+m_j}} \frac{z^{m_1+\ldots+m_{j+1}}}{m_1! \cdots m_{j+1}!}. \tag{B.1}$$

For $a = 0, b = 1, z = -1/2\sigma^2$ it is clear that in the case of small $1/2\sigma^2$ the derivatives all tend to 0 for $j \geq 1$. This is so, because the lowest order in $1/2\sigma^2$ in the power series is of order 1 which together with the prefactor $(2\sigma^2)^{-j}$ clearly approaches zero as $1/2\sigma^2$ goes to zero.

In contrast to the first cumulant the higher order rescaled cumulants do not show any logarithmic divergence but rather asymptotic behaviour given by a non-zero constant and zero. It is important that in this case $1/2\sigma^2$ was always assumed to be independent of $M$ such that the only $M$ contribution to the cumulants is the prefactor $M/2$.

# C Approximation of the distribution

This conclusion is consistent with an approximation of the characteristic function and subsequent calculation of the distribution. Importantly, the characteristic function has significant values for $k$ close to zero only, see Fig. 12.

Splitting off the Gaussian contribution of the characteristic function

$$\chi_S(k|M^\alpha) = \exp\left(\kappa_1 \iota k + \frac{\kappa_2}{2}(\iota k)^2\right)\exp\left(\sum_{j=3}^{\infty}\frac{\kappa_j}{j!}(\iota k)^j\right), \tag{C.1}$$

which are the only significant contributions for $k$ close to zero. This means that it is sufficient to consider powers of $\iota k$ up to two in the argument of the exponential function and powers up to five linearly. Thus, the exponential function containing those higher powers is approximated by

$$\exp\left(\sum_{j=3}^{\infty}\frac{\kappa_j}{j!}(\iota k)^j\right) \approx 1 + \sum_{j=3}^{4}\frac{\kappa_j}{j!}(\iota k)^j . \tag{C.2}$$

In principle even higher orders may be taken into account. However, the expression will become lengthy due to contributions combining lower order terms. From the above considerations we expect that these higher order will drop off with large $M$. We confirm this in the following and therefore they neglect these terms here.

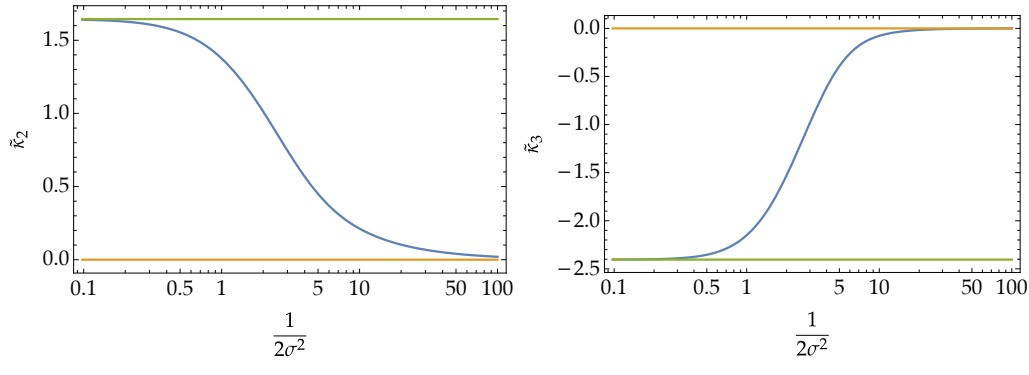

(a) Rescaled second cumulant $\tilde{\kappa}_2$.    (b) Rescaled third cumulant $\tilde{\kappa}_3$.

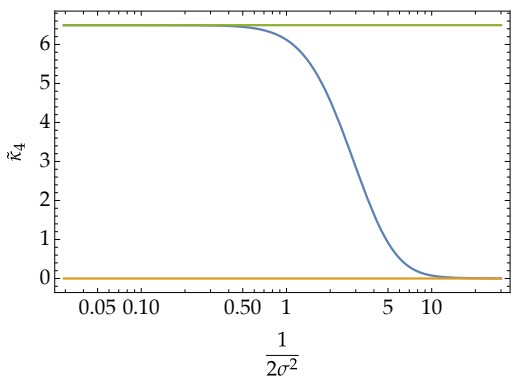

(c) Rescaled fourth cumulant $\tilde{\kappa}_4$.

Figure 11: Comparison of the rescaled (a) second, (b) third, (c) fourth cumulant, blue line, the asymptotics (a) $\psi_1(1)$, (b) $\psi_2(1)$, (c) $\psi_3(1)$, green line, and zero, orange line, versus $1/2\sigma^2$.

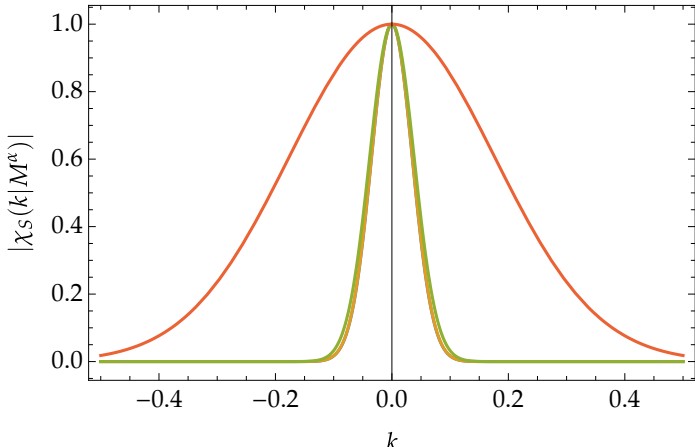

Figure 12: Comparison of $|\chi_S(k|M^\alpha)|$ versus $k$ for $M = 1000$ and $\alpha = -1$ (blue), $\alpha = -0.5$ (orange), $\alpha = 0$ (green) and $\alpha = 0.5$ (red).

We now carry out the Fourier transformation 35 and arrive at the intermediate result

$$p_S(F|M^\alpha) \approx \left(1 + \sum_{j=3}^{4} \frac{\kappa_j}{j!}(-\partial_F)^j\right)\frac{1}{\sqrt{2\pi\kappa_2}}\exp\left(-\frac{(F-\kappa_1)^2}{2\kappa_2}\right).$$ (C.3)

From the definition of the Hermite polynomials [20] it follows that the distribution is of the form

$$p_S(F|M^\alpha) \approx \frac{1}{\sqrt{2\pi\kappa_2}}\exp\left(-\frac{(F-\kappa_1)^2}{2\kappa_2}\right)\left(1 + \sum_{j=3}^{4}\frac{\kappa_j}{j!\sqrt{2\kappa_2}^j}H_j\left(\frac{(F-\kappa_1)}{\sqrt{2\kappa_2}}\right)\right),$$ (C.4)

which is Gaussian in leading order with additional polynomial corrections. Importantly the deviations around the Gaussian behaviour are weighted by the ratio in Eq. (84) which shows that the quantity indeed measures the relevance of certain orders. This ratio appears linearly instead of in an exponential because all other terms inside the exponential function were neglected and these higher order terms drop off faster.

Furthermore, the contribution of higher orders are always measured on the scale of the standard deviation. Even if higher order cumulants are non-zero, or grow with $M$, the distribution will still tend to a Gaussian, as is to be expected from the central limit theorem. However, this makes it possible to fit a distribution which reproduces said higher order cumulants while also fulfilling the Gaussian limit. This allows us to determine whether the fit accurately captures the behaviour of the distribution $p_S(F|M^\alpha)$ in a given interval for $M$ or if it only obeys the same Gaussian limit.

The approximation of the characteristic function for small $k$ is equivalent to the large $M$ approximation of the distribution, because higher powers of $k$ can be expressed by derivatives with respect to $F$ which then in turn yield the prefactors in Eq. (84). The different powers in $k$ correspond to the contributions from different orders of the cumulants such that if one takes only very small $k$ into account the higher order cumulants are negligible and the dependence of the interval of significant $k$-values on $M$ is shown in Fig. 12.

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
