# Peer review of "Random Matrices and the Free Energy of Ising-Like Models with Disorder"

_SciPost Physics, doi:SciPost Phys. 17, 122 (2024)_

## Round 1 · Referee Report · Anonymous (Referee 1) · 2024-4-16

Strengths

1 - extensive and detailed derivations
2 - clear motivation

Weaknesses

1 - figure layout not optimal in some places

Report

This work provides a clear and systematic approach to studying the emergence of log-normal distributions of the free energy for large disordered Ising systems and their asymptotic Gaussian form. Overall, the arguments provided are plausible and can be followed well. The manuscript is well organized, starting with a clear motivation, defining a specific problem, and solving it step-by-step. The high level of detail allows one to follow the derivations, ask critical questions (see below), and enable future work to reproduce (not done here) and build upon the present results.

For my report, I attempted to closely follow the derivations (and apologize for the extra time this took) to formulate the following comments that the authors may want to consider.

  • I think the abstract should reflect more on the content of the paper. Large efforts are invested in the analytical derivations towards the characteristic function that is later "merely" numerically evaluated for the finite-size corrections to the scaling arguments for the limit of M->infty. I was initially under the impression that the work involved numerical simulations after reading "We show numerically [that] [...] and in the limit of infinitely large matrices the distributions become Gaussian".

  • In Eq. (1) I fail to see why an additional variable $K_{ij}$ was introduced

  • Given the high level of detail and its later importance, I was hoping for more details on the origin of $Q$ and $\kappa$.

  • Below Eq. (23) there is a distinction between $(0)_0$ and $(0)_n$ but both are equal to 1, such that it could also be valid for $n\geq 0$.

  • Below Eq. (46) it is noted that all eigenvalues of Q being on a unit circle is a particularly interesting case, but I thought this is by construction. Since the final part of Sec. 3 is so fundamental, I could imagine that making this crystal clear may be beneficial. Also, it is picked up in the last paragraph again.

  • Eq. (59) seems odd because the first equation is identical up to the indices $\xi$ and $\xi +1$.

  • p. 15 :The statement "logarithmic growth [...] with additional linear contributions" irritates me. It rather looks like linear growth (M/2) with logarithmic contributions (MlogM) but maybe I am missing sth.

  • Fig. 1: which $\alpha$ is shown here? If this is tested numerically, it should be verified for different values of $\alpha$, especially since Eq. (74) is super explicit about the $\alpha$ dependence.

  • Same question regarding the choice of $\alpha$ applies to the following figures.

  • p. 16: When it is "plausible that Eq. 75 also holds for j>4" then the statement "Thus, the leading order of all higher order cumulants is always..." appears a bit strong. Maybe the word "implies that" would work better

  • Below Eq.(79) it is stated that the equation is independent of $\alpha$ but it explicitly involves an $\alpha$. Of course, the limit being zero becomes independent of $\alpha$ but the exponential decay does depend on $\alpha$.

  • Eq. (81) should be only valid for $j>1$

  • p. 19 in the synopsis, I do not understand the statement that the leading order behavior is not dependent on $\alpha$ if there is a logarithmic behavior that depends on $\alpha$ for negative values. However, as stated above I believe the logarithmic behavior is a correction, or not?

  • Fig. 5: One cannot see the x labels at all. As a suggestion, it would be much more comprehensible for me if the absolute relative error was plotted on a log scale. The same applies to Fig. 6

  • p. 21 "Importantly, the plots show that the qualitative behavior is captured ..." -> I would argue that small relative errors imply that also the quantitative behavior is captured, or am I mistaking sth?

  • p. 22: It was unclear to me at this point where "the numerical data" really came from.

  • Table 1: I would suggest to start with the leading-order term. Otherwise, I was confused about the sign change for $\alpha=-1/4$ or the statement that "For all $\alpha>0$ the leading order of $\sigma$ will no longer be proportional to $M^{-1/2}$"

  • Fig. 7a: Is this really $\kappa_1$ (figure axis) or $\kappa_3$ (caption)?

  • Fig. 8/9: Why is there no crossover between log-normal and Gaussian approximation in sight? For large $M$ the Gaussian approximation should become much better, right?

Recommendation

Ask for minor revision

---

## Round 1 · Referee Report · Anonymous (Referee 2) · 2024-4-17

Strengths

1 - Derivations very clear and easy to follow
2 - Paper is generally well written
3 - Topic is interesting and timely

Weaknesses

1 - The construction of the model seems a bit 'artificial' at places
2 - Sometimes the mathematical derivations conceal or cloud the physical interpretation (or at least sometimes it would be desirable to better weave physical considerations on the Ising model with surface disorder into the mathematical analysis)

Report

I enjoyed reading this paper, which is exceedingly well written and easy to follow. I have a few minor comments and requests for clarifications.

1 - Pag. 3 'elude' should probably be 'allude' 2 - An explicit expression for the matrix Q defined in Eq. 3 is never provided. It is true that it is essentially proven irrelevant in the following (at least as long as its eigenvalues are exactly on the unit circle), but it is somehow unsatisfactory that only its features (real skew-circulant etc) are provided, but not its explicit expression 3 - As the paper is written to be as self-contained and pedagogical as possible, I would cite more recent pedagogical introductions to RMT alongside Ref. [11] on pag. 4, for instance Potters' 'A first course in Random Matrix Theory' and Livan's et al. 'Introduction to Random Matrices: theory and practice' 4 - Section 2.3 would benefit from an explicit example (say, a 6x6 matrix) to show the structure of a general skew-circulant matrix. At the moment, it is difficult to picture in one's mind how such matrices look like from the definition in Eq. 12 5 - In Section 3, the reason why this precise choice of the disorder (skew-circulant) is made is unclear. Perhaps one should add that this choice simplifies (or makes it at all possible) a sounder analytical treatment, and briefly explain why it is so. Also, it is not completely clear to me what this choice actually means in terms of 'surface spins' - if I were to simulate this model on a computer, how should the spins $\pm 1$ on the outer interface be drawn to precisely reproduce this skew-circulant model? 6 - After eq. 54 'deriviated' (typo) ---> differentiated 7 - The first sentence on pag. 12 sounds quite odd to me. Why should it be 'remarkable' that the first cumulant is an incomplete Gamma function? Is this very expected or highly unexpected for some reason? Also, how does this exact cumulant compare with the first cumulant of a log-normal distribution, which (I believe) is claimed to provide a good approximation for the full distribution of the free energy? 8 - Before eq. 64, it is again not clear to me on what grounds the coupling strength is scaled in this way with the system size. Since much of the rest of the paper is related to the mathematical analysis of different cases depending on $\alpha$, it would seem slightly unsatisfactory to leave the reader wonder whether this is simply an ad hoc choice, or if instead there are deeper physical reasons for assuming this scaling. 9 - All the remaining discussion is mathematically sound and interesting, but I would suggest tying in the mathematical results a bit better with some physical insight on the Ising physics. One has the feeling at places that the authors have been slightly 'carried away' in their desire to show how much could be done analytically on this model, at the expenses of a clearer connection to the physical model they ought to describe. 10 - In the Conclusions, there are two sentences very close to each other, where first the distribution is stated to tend to a Gaussian, and immediately afterwards it is stated to be log-normal. A quick reader may find this slightly contradictory. I would urge the authors to specify a bit better which cases are considered and under which limits, yielding to Gaussian or lognormal (but clearly not both simultaneously).

Requested changes

See above

Recommendation

Ask for minor revision

---

## Round 2 · Referee Report · Anonymous (Referee 1) · 2024-7-17

Report

The revised version and the authors reply addresses all of my remarks appropriately. I have one last remark that in the revision, the colors in Fig. 8 and Fig. 9 changed but the association between colors and models in the caption remained. Maybe this was fixing a previous mistake, in which case it would be fine, but it rather appears as a mistake in the remake of the figure (which now includes additional lines with power-law decays that are not mentioned in the discussion of the figure) because the Gaussian similarity measure is not the upper bound anymore.

Recommendation

Publish (meets expectations and criteria for this Journal)

  • validity: -
  • significance: -
  • originality: -
  • clarity: -
  • formatting: -
  • grammar: -

Author:  Nils Gluth  on 2024-07-23  [id 4644]

(in reply to Report 1 on 2024-07-17)

Dear Referee,

we are very grateful for spotting the change in colors in figure 8 and 9. We changed the colors but forgot to change the caption accordingly. The colors should be blue for Gaussian, orange for the fourth order approximation and green for the log-normal fit such that the figures convey the same message as in the prior version of the manuscript. We added the additional power-law decays such that the difference between positive and negative $\alpha$ is visually clearer.

With best regards

Nils Gluth

---

## Round 2 · Referee Report · Anonymous (Referee 2) · 2024-7-29

Report

All my previous remarks have been implemented with good care and I am satisfied that the paper can now be published.

Recommendation

Publish (easily meets expectations and criteria for this Journal; among top 50%)

---

## Round 2 · Author Response

Dear Mr Beenakker,

We have resubmitted the revised version of the manuscript.
We want to express our gratitude to the two referees, and we think that their insightful comments have improved the quality of our paper.
Please find a list of the changes made in accordance with the referees comments below.

With best regards

Nils Gluth

---

## Round 2 · List of Changes

Our list of changes contains the referees suggestion and our implementation in the manuscript which we separate by a paragraph.

Referee 1 1) I think the abstract should reflect more on the content of the paper. Large efforts are invested in the analytical derivations towards the characteristic function that is later "merely" numerically evaluated for the finite-size corrections to the scaling arguments for the limit of $M\to \infty$. I was initially under the impression that the work involved numerical simulations after reading "We show numerically [that] [...] and in the limit of infinitely large matrices the distributions become Gaussian".

We changed the two relevant sentences in the abstract to: "We chose skew-circulant random matrices and analytically compute the characteristic function of the free energy distribution. From the characteristic function we numerically calculate the distribution and show that it becomes log-normal for sufficiently large dimensions of the disorder matrices, and in the limit of infinitely large matrices tends to a Gaussian."

2) In Eq. (1) I fail to see why an additional variable $K_{ij}$ was introduced.

We now use the usual coupling constant $J$ and added the inverse temperature $\beta$ to the latter equations and put an emphasis on the fact that we use the dimensionless free energy in our model.

3) Given the high level of detail and its later importance, I was hoping for more details on the origin of $Q$ and $\kappa$.

We modified the first part of Sec. 2.1 to be more specific. We rewrote the Hamiltonian to explictly be anisotropic with only two couplings $J^{\leftrightarrow}, J^{\updownarrow}$ and the system dimensions $L$ and $M$. This allows us to give an explicit expression for $Q$ in the so-called Hamiltonian limit in Eq. (5).

4) Below Eq. (23) there is a distinction between $(0)_0$ and $(0)_n$ but both are equal to 1, such that it could also be valid for $n\geq0$.

We replaced our definition of the Pochhammer symbol via Gamma functions by a definition via a product.

5) Below Eq. (46) it is noted that all eigenvalues of $Q$ being on a unit circle is a particularly interesting case, but I thought this is by construction. Since the final part of Sec. 3 is so fundamental, I could imagine that making this crystal clear may be beneficial. Also, it is picked up in the last paragraph again.

We replaced the assumption of a general $Q$ in Sec. 3 and calculated the characteristic function for a $Q$ with the same properties as in Sec. 2.1. We changed the last paragraph of Sec. 3 to "As shown, the characteristic function does not depend on Q since all eigenvalues of Q are on the unit circle and therefore have radius one. We want to mention that it is also possible to calculate the characteristic function when Q has arbitrary complex eigenvalues."

6) Eq. (59) seems odd because the first equation is identical up to the indices $\xi$ and $\xi+1$.

We removed the second equality to avoid confusion, which we had added to clarify the connection to the definition of the polygamma-function.

7) p. 15 :The statement "logarithmic growth [...] with additional linear contributions" irritates me. It rather looks like linear growth (M/2) with logarithmic contributions (MlogM) but maybe I am missing sth.

We changed our original statement which was indeed confusing and replaced it with "From the above asymptotics we conclude that an important characteristic of the case $\alpha<0$ is the leading term of the first cumulant, the mean value, is given by $M \ln M$ and we have additional linear contributions."

8) Fig. 1: which $\alpha$ is shown here? If this is tested numerically, it should be verified for different values of $\alpha$, especially since Eq. (74) is super explicit about the $\alpha$ dependence.

We added the sentences "We adress the scaling behaviour of the higher order cumulants numerically. We do this by analysing the rescaled cumulants $\tilde{\kappa}_j$ which only depend on $M^\alpha$, as seen in Eq. (60) for $1/(2\sigma^2) = M^\alpha$, and therefore we can plot them as functions of $M^\alpha$. For $\alpha<0$ we know that $M^\alpha \in (0,1]$ and recall that the second cumulant, the variance, converges to $\psi_1(1)M/2 = M\pi^2/12$ for small $1/(2\sigma^2)$, see Appendix B." to clarify our fitting procedure.

9) Same question regarding the choice of $\alpha$ applies to the following figures.

We refer to the reply to the prior comment.

10) p. 16: When it is "plausible that Eq. 75 also holds for $j>4$" then the statement "Thus, the leading order of all higher order cumulants is always..." appears a bit strong. Maybe the word "implies that" would work better

We have implemented the referees suggestion and changed the sentence to "This implies that the leading order of all higher order cumulants is always linear in M for $\alpha< 0$.".

11) Below Eq.(79) it is stated that the equation is independent of $\alpha$ but it explicitly involves an $\alpha$. Of course, the limit being zero becomes independent of $\alpha$ but the exponential decay does depend on $\alpha$.

Our formulation was unclear, we changed it to "for all $\alpha>0$".

12) Eq. (81) should be only valid for $j>1$

We have clarified this by modifying the sentence to "The behaviour of the higher order cumulants is given by...".

13) p. 19 in the synopsis, I do not understand the statement that the leading order behavior is not dependent on $\alpha$ if there is a logarithmic behavior that depends on $\alpha$ for negative values. However, as stated above I believe the logarithmic behavior is a correction, or not?

We changed the formulation in the synopsis to hopefully be more precise by writing "It is interesting that the leading order behaviour is not dependent on the value of $\alpha$, except for the first cumulant $\kappa_1$." For the logarithmic behaviour we refer to the response to the prior comment.

14) Fig. 5: One cannot see the x labels at all. As a suggestion, it would be much more comprehensible for me if the absolute relative error was plotted on a log scale. The same applies to Fig. 6.

We implemented the referees suggestion and both figures are now on a log scale, we plot the absolute value since not all of the relative errors share a common sign.

15) p. 21 "Importantly, the plots show that the qualitative behavior is captured ..." $\rightarrow$ I would argue that small relative errors imply that also the quantitative behavior is captured, or am I mistaking sth?

We agree with the referee and changed "qualitative" to "quantitative".

16) p. 22: It was unclear to me at this point where "the numerical data" really came from.

We added the sentence "We numerically calculated the distribution with Mathematica ..." to make it clear that we produced the numerical data.

17) Table 1: I would suggest to start with the leading-order term. Otherwise, I was confused about the sign change for $\alpha=-1/4$ or the statement that "For all $\alpha>0$ the leading order of $\sigma$ will no longer be proportional to $M^{-1/2}$"

We implemented the referees suggestion and switched the order of the terms.

18) Fig. 7a: Is this really $\kappa_1$ (figure axis) or $\kappa_3$ (caption)?

We have changed $\tilde{\kappa}_3$ to $\tilde{\kappa}_1$ in the caption to fit with the figure, which had the correct label.

19) Fig. 8/9: Why is there no crossover between log-normal and Gaussian approximation in sight? For large M the Gaussian approximation should become much better, right?

We added the small paragraph "The Gaussian provides an upper bound since we know from the central limit theorem that for $M\to\infty$ our distribution is Gaussian. However, we are particularly in the behaviour deviating from the Gaussian behaviour and therefore for a reasonable fit we expect that it not only decays with $M$ but also that it is below the Gaussian curve and only meets it for $M\to\infty$." at the end of the chapter discussing this question.

Referee 2 1) Pag. 3 'elude' should probably be 'allude'

We replaced the word "elude" by "allude".

2) An explicit expression for the matrix Q defined in Eq. 3 is never provided. It is true that it is essentially proven irrelevant in the following (at least as long as its eigenvalues are exactly on the unit circle), but it is somehow unsatisfactory that only its features (real skew-circulant etc) are provided, but not its explicit expression.

We added an explicit expression for $Q$, see reply to the third point of the first referee.

3) As the paper is written to be as self-contained and pedagogical as possible, I would cite more recent pedagogical introductions to RMT alongside Ref. [11] on pag. 4, for instance Potters' 'A first course in Random Matrix Theory' and Livan's et al. 'Introduction to Random Matrices: theory and practice'

We added the mentioned works to our citations.

4) Section 2.3 would benefit from an explicit example (say, a 6x6 matrix) to show the structure of a general skew-circulant matrix. At the moment, it is difficult to picture in one's mind how such matrices look like from the definition in Eq. 12

We added eq. (14) which contains an explicit example for $M=4$ to visualise the abstract definition of skew-circulant matrices.

5) In Section 3, the reason why this precise choice of the disorder (skew-circulant) is made is unclear. Perhaps one should add that this choice simplifies (or makes it at all possible) a sounder analytical treatment, and briefly explain why it is so. Also, it is not completely clear to me what this choice actually means in terms of 'surface spins' - if I were to simulate this model on a computer, how should the spins ±1 on the outer interface be drawn to precisely reproduce this skew-circulant model?

We added "This choice constitutes the simplest choice since we discussed in Sec. 2.3 all skew-circulant matrices share a common eigenbasis and therefore $Q$ and $S$ can be diagonalised simultaneously. Gaussian distributed skew-circulant type of disorder is realised by Gaussian distributed spin orientations in the Fourier space. In the real space the skew-circulant nature would manifest through interaction of all $M$ spins on the surface. However, the interaction only depends on the distance between the two spins." to make the choice clearer and include the realisation of such a type of disorder.

6) After eq. 54 'deriviated' (typo) $\rightarrow$ differentiated

We replaced the word "deriviated" by "differentiated".

7) The first sentence on pag. 12 sounds quite odd to me. Why should it be 'remarkable' that the first cumulant is an incomplete Gamma function? Is this very expected or highly unexpected for some reason? Also, how does this exact cumulant compare with the first cumulant of a log-normal distribution, which (I believe) is claimed to provide a good approximation for the full distribution of the free energy?

We changed our formulation to be "This is a remarkably simple expression since ..." which reflects that we think that compared to the characteristic function and all higher order cumulants the first cumulant is very simple. Furthermore, we added the paragraph "However, for positive $\alpha$ there are deviations for the mean which is to be expected since we fixed $s = \kappa_1$ and the first cumulant of a log-normal distribution is given by $\kappa_{\text{log},1} = f_0(1 - \exp(\sigma^2/2)) + s$. Similarly, the higher order cumulants show larger deviations as well, nevertheless, the qualitative behaviour is captured by the fit, see Fig. 7. We generally see that for the considered values of $\alpha$ the case $\alpha = 1/4$ performs the worst, in the range of $M$ values chosen here. We suspect that the smaller the $\alpha > 0$ the higher the $M$ values would have to be to achieve results comparable to those of negative $\alpha$. A plausible reason for this is the behaviour of the mean $\kappa_1 \sim M \exp(-M^\alpha)$ which grows linearly and then decays exponentially with growing $M$, the scale on which this decay occurs is determined by $\alpha$ and the smaller $\alpha$ the larger $M$ has to be." at the end of Sec. 6.2.2. to further supplement the discussion of the mean value and the difficulty of achieving a satisfying fit.

8) Before eq. 64, it is again not clear to me on what grounds the coupling strength is scaled in this way with the system size. Since much of the rest of the paper is related to the mathematical analysis of different cases depending on $\alpha$, it would seem slightly unsatisfactory to leave the reader wonder whether this is simply an ad hoc choice, or if instead there are deeper physical reasons for assuming this scaling.

We added the two sentences "This choice is motivated by models like the famous Sherrington-Kirkpatrick-model in which the coupling strength also depends on the size of the system [24]. We leave the exponent variable to investigate differences in the behaviour depending on the choice of the exponent." to motivated our choice for the coupling strength.

9) All the remaining discussion is mathematically sound and interesting, but I would suggest tying in the mathematical results a bit better with some physical insight on the Ising physics. One has the feeling at places that the authors have been slightly 'carried away' in their desire to show how much could be done analytically on this model, at the expenses of a clearer connection to the physical model they ought to describe.

We hope that the implemented changes have drawn a clearer connection to the physical motivation behind the research. First and foremost it was our goal to study the distribution of log-determinants of matrices $Q+\kappa$ and chose the most accessible path which unfortunately is not a studied model, to our knowledge. However, it is interesting that we still find the same behaviour.

10) In the Conclusions, there are two sentences very close to each other, where first the distribution is stated to tend to a Gaussian, and immediately afterwards it is stated to be log-normal. A quick reader may find this slightly contradictory. I would urge the authors to specify a bit better which cases are considered and under which limits, yielding to Gaussian or lognormal (but clearly not both simultaneously).

We clarified this by reformulating the sentence to "For the limit $M\to\infty$ we found that the central limit theorem holds and the distribution tends to a becomes Gaussian with mean $\kappa_1$ and variance $\kappa_2$." to better distinguish between the intermediate limit of the log-normal distribution and the $M\to\infty$ limit in which the distribution becomes Gaussian.

---

## Editorial Decision

published